# Do PAC-Learners Learn the Marginal Distribution?

**Max Hopkins**                                                                    MH4067@PRINCETON.EDU
*Princeton University*

**Daniel M. Kane**                                                                 DAKANE@CS.UCSD.EDU
*University of California, San Diego*

**Shachar Lovett**                                                                 SLOVETT@UCSD.EDU
*University of California, San Diego*

**Gaurav Mahajan**                                                             GAURAV.MAHAJAN@YALE.EDU
*Yale University*

**Editors:** Gautam Kamath and Po-Ling Loh

## Abstract

The Fundamental Theorem of PAC Learning asserts that learnability of a concept class $H$ is equivalent to the *uniform convergence* of empirical error in $H$ to its mean, or equivalently, to the problem of *density estimation*, learnability of the underlying marginal distribution with respect to events in $H$. This seminal equivalence relies strongly on PAC learning's 'distribution-free' assumption, that the adversary may choose any marginal distribution over data. Unfortunately, the distribution-free model is known to be overly adversarial in practice, failing to predict the success of modern machine learning algorithms, but without the Fundamental Theorem our theoretical understanding of learning under distributional constraints remains highly limited.

In this work, we revisit the connection between PAC learning, uniform convergence, and density estimation beyond the distribution-free setting when the adversary is restricted to choosing a marginal distribution from a known family $\mathscr{P}$. We prove that while the traditional Fundamental Theorem fails, a finer-grained connection between the three fundamental notions continues to hold:

1. PAC-Learning is strictly sandwiched between two relaxed models of density estimation, differing only in whether the learner knows the set of well-estimated events in $\mathcal{H}$.

2. Under reasonable assumptions on $H$ and $\mathscr{P}$, density estimation is equivalent to *uniform estimation*, a weakening of uniform convergence allowing non-empirical estimators.

Together, our results give a clearer picture of how the Fundamental Theorem extends beyond the distribution-free setting and shed new light on the classically challenging problem of learning under arbitrary distributional assumptions.

**Keywords:** PAC-Learning, Density Estimation, Uniform Convergence

## 1. Introduction

Valiant (1984) and Vapnik and Chervonenkis (1974) Probably Approximately Correct (PAC)-Learning is the foundation of modern learning theory. Traditionally, the model captures learnability of a hypothesis class $(X, H)$, a set $X$ of 'raw data' and a family of binary classifiers $H$, against a *worst-case* or '*distribution-free*' adversary, meaning algorithmic success is always measured against the worst possible marginal distribution over data. This setting gives rise to an elegant theory of learning centered around the 'Fundamental Theorem of PAC Learning', stating the complexity of learning $H$ is controlled by *uniform convergence*, the extent to which the empirical mean of a large enough

sample uniformly estimates the error of every $h \in H$ (Vapnik and Chervonenkis, 1974; Blumer et al., 1989; Haussler, 1992).

Unfortunately, in practice, distribution-free PAC-learning is overly adversarial. Traditional lower bounds rely on contrived data distributions not seen in practice, and modern architectures like deep nets learn much faster than PAC theory predicts (Neyshabur et al., 2014; Zhang et al., 2021; Nagarajan and Kolter, 2019). At a theoretical level, this raises a natural question:

*Can we characterize learnability under a **restricted** adversary?*

We study a foundational variant of this question called the *distribution-family model*, which restricts the adversary to choosing data from a known family $\mathscr{P}$ of marginal distributions. Characterizing learnability in this setting is an old and challenging problem (Vapnik and Chervonenkis, 1974, 1971; Benedek and Itai, 1991; Natarajan, 1992; Dudley et al., 1994), dating back to work of Benedek and Itai (1991) observing the failure of uniform convergence in the setting. Since that time, progress on the distribution family model has been largely negative, with Dudley et al. (1994) falsifying a conjectured characterization of Benedek and Itai (1991) based on metric entropy, and recent work of Lechner et al. (2023) showing the model has no scale-insensitive combinatorial characterization.

In this work, we revisit the conventional knowledge that the Fundamental Theorem fails beyond the distribution-free setting and aim to offer a more nuanced view of the connection between uniform convergence and its variants and learning under distributional assumptions. Toward this end, it is constructive to consider a rephrasing of the Fundamental Theorem in terms of a standard statistical framework: *density estimation*. Given a triple $(\mathscr{P}, X, H)$ and oracle access to a distribution $D \in \mathscr{P}$, the density estimation problem asks the learner to output $D' \in \mathscr{P}$ that is close to $D$ in the total variation metric with respect to $H$:[1]

$$TV_{H \Delta H}(D, D') := \max_{S \in H \Delta H}(|D(S) - D'(S)|).$$

This is natural in the context of supervised learning where events in $H \Delta H$ measure the error between hypotheses, and is clearly closely related to uniform convergence. Indeed, the following folklore rephrasing (Yatracos, 1985; Devroye and Lugosi, 2001; Chan et al., 2014; Ashtiani et al., 2018) of the Fundamental Theorem states that density estimation, uniform convergence, and PAC learning are all equivalent in the distribution-free model.

**Theorem 1 (The Fundamental Theorem of PAC Learning (Informal))** *In the distribution-free setting:*

*Density Estimation = PAC-Learning = Uniform Convergence.*

While Benedek and Itai (1991) observe the latter equivalence breaks in the distribution-family model, their example does not contradict a potential equivalence with density estimation. Indeed, density estimation is a common strategy in learning with distributional assumptions (see e.g. Blanc et al. (2023)). We will work with the following weakening (equivalent in the distribution free setting), which is clearly sufficient to learn so long as each $(D, X, H)$ in $(\mathscr{P}, X, H)$ is individually learnable.[2]

---

1. Here $H \Delta H := \{h \Delta h' : h, h' \in H\}$ denotes the set of symmetric differences in $H$.

2. Formally here we mean the $(D, X, H)$ are learnable in a uniformly bounded number of samples, which Benedek and Itai (1991) proved is equivalent to $(\mathscr{P}, X, H)$ having uniformly bounded metric entropy (see Definition 12). If this does not hold $(\mathscr{P}, X, H)$ is trivially unlearnable, and the problem becomes uninteresting. Throughout this work, we will always assume this condition.

**Definition 2 (Intermediate Density Estimation (Informal Definition 14))** $(\mathscr{P}, X, H)$ *has an intermediate density estimator (IDE) if there exist a learner and (sample dependent) subset* $G \subset H$ *satisfying:*

1. *The learner accurately estimates* $d_D(g, g')$ *for all* $g, g' \in G$

2. *Every fixed* $h \in H$ *lies in* $G$ *with high probability*

3. *The learner knows* $G$.

In other words, while standard density estimation requires outputting a good estimate on the measure of all elements in $H \Delta H$ simultaneously, intermediate estimation only requires this for $G \Delta G$, where $G$ is known and 'typically' contains any fixed hypothesis in the class.

On its surface, necessity and sufficiency of intermediate estimation seems reasonable. It holds in the distribution-free setting, and it is clear that any PAC learner must infer *some* information about the measure of $H \Delta H$, or it would not be able to distinguish between good and bad hypotheses. Unfortunately, it turns out such a direct extension fails: even the relaxed intermediate density estimation is not necessary for PAC learning with arbitrary distribution families.

**Proposition 3 (Intermediate Estimation Fails Necessity (Informal Theorem 17))** *There exists a PAC-learnable class* $(\mathscr{P}, X, H)$ *with no intermediate density estimator.*

On the other hand, following our prior intuition, there should be some relaxation of density estimation that is necessary for PAC learning. One natural weakening is to drop the condition that the learner knows the set of well-estimated events. We call this notion weak density estimation:

**Definition 4 (Weak Density Estimation (Informal Definition 15))** $(\mathscr{P}, X, H)$ *has a weak density estimator (WDE) if there exists a learner and a (sample-and-distribution dependent) subset* $G \subset H$ *satisfying:*

1. *The learner accurately estimates* $d_D(g, g')$ *for all* $g, g' \in G$

2. *Every fixed* $h \in H$ *lies in* $G$ *with high probability.*

Unlike the intermediate model, we prove weak estimation *is* necessary to PAC-learn (see Proposition 21). One might then hope the condition also remains sufficient, but this turns out to be false in general as well.

**Proposition 5 (Weak Density Estimation Fails Sufficiency (Informal Theorem 22))** *There exists a class* $(\mathscr{P}, X, H)$ *with a weak density estimator that is not PAC-learnable.*

Taken together, these results lead to our main theorem: PAC-Learning with distributional assumptions is strictly sandwiched between two close models of density estimation.

**Theorem 6 (The PAC-Density Sandwich Theorem)** *In the distribution-family setting:*

$$\text{Weak Density Estimation} \subsetneq \textbf{PAC-Learning} \subsetneq \text{Intermediate Density Estimation}$$

Finally, we return to the topic of uniform convergence and its relation to density estimation and learning. While uniform convergence is strictly stronger than learning and density estimation in the distribution family model, a natural relaxation called *uniform estimation* which allows for non empirical estimators is more closely related. Indeed, we prove that when $(\mathscr{P}, X, H)$ has uniformly bounded metric entropy, the two notions are equivalent.

**Proposition 7 (Density Estimation $\iff$ Uniform Estimation (Informal Theorem 27))** $(\mathscr{P}, X, H)$ *has an (intermediate) density estimator if and only if it has an (intermediate) uniform estimator.*

We emphasize Proposition 7 only holds under the assumption that each $(D, X, H)$ in the class is learnable in a uniformly bounded number of samples. Indeed the statement is trivially false otherwise, see Section 6 for further discussion.

We remark that while this connection is standard in the distribution-free setting (Yatracos, 1985), the argument in the distribution-family model is more subtle and combines standard methods in density estimation (Yatracos, 1985; Devroye and Lugosi, 2001) with recent connections between covering and learning with distributional assumptions (Hopkins et al., 2024). We conjecture that Theorem 7 should actually fail for the corresponding 'weak' estimation variant. Here weak uniform estimation (estimating error for most hypotheses in the class without knowing the set of 'good' estimates) is immediate from Chernoff. On the other hand, weak density estimation asks not just for a good estimate of most $d_D(h, h')$, but rather for a large set $G$ where *every* distance $d_D(g, g')$ is well estimated. This is a more involved condition, and we leave a characterization of classes satisfying the property as an open problem for future work.

Taken together, Theorem 6 and Proposition 7 provide a finer-grained picture of the classical connection between supervised learning, uniform convergence, and density estimation. It remains an interesting open problem to prove further positive results for learning with distributional assumptions as well. One promising direction is to ask whether there are natural assumptions on the class $\mathscr{P}$ over which uniform estimation and density estimation are necessary. Another interesting direction is to explore classical paradigms such as sample compression or stability, known to characterize learnability in other settings where uniform convergence fails (Shalev-Shwartz et al., 2010; David et al., 2016).

## 2. Background and Related Work

### 2.1. PAC Learning, Uniform Convergence, and Metric Entropy

Let $X$ be a set, $H$ a family of binary classifiers over $X$, and $\mathscr{P}$ a family of distributions over $X$.

**Definition 8 (Distribution Family PAC-learning)** *We say $(\mathscr{P}, X, H)$ is PAC-learnable if there exists a function $n = n_{PAC}(\varepsilon, \delta)$ such that for all $\varepsilon, \delta > 0$, there is an algorithm $A$ which for every $D \in \mathscr{P}$, and $h \in H$ satisfies:*

$$\Pr_{S \sim (D \times h)^n}[d_D(A(S), h) > \varepsilon] \leq \delta,$$

*where $d_D(h, h') = \Pr_{x \sim D}[h(x) \neq h'(x)]$ is the classification distance.*

In general one might also consider an *agnostic* model allowing arbitrary labelings, but this is statistically equivalent to the above (Hopkins et al., 2024) so we focus on the simpler realizable case.

Characterizations of the distribution-family model are known in several special cases. Most famously, the 'distribution-free' setting where $\mathscr{P}$ consists of the set of all distributions is characterized by uniform convergence (Vapnik and Chervonenkis, 1974; Blumer et al., 1989).

**Definition 9 (Uniform Convergence)** *We say $(\mathscr{P}, X, H)$ satisfies the uniform convergence property if there exists a function $n = n_{UC}(\varepsilon, \delta)$ such that for all $\varepsilon, \delta > 0$, $D \in \mathscr{P}$, and $h \in H$, the*

*empirical error of every $h' \in H$ approaches its true error uniformly:*

$$\Pr_{S \sim (D \times h)^n}[\exists h' \in H : |err_S(h') - err_{D \times h}(h')| > \varepsilon] \leq \delta,$$

*where $err_S$ and $err_{D \times h}$ are the empirical and true error respectively:*

$$err_S(h') = \frac{1}{|S|} \sum_{(x,y) \in S} \mathbf{1}[h'(x) \neq y], \quad err_{D \times h}(h') = \Pr_{(x,y) \sim D \times h}[h'(x) \neq y].$$

Bounds on uniform convergence or estimation in the distribution-free case typically depend on a complexity measure of the class called the *growth function* $\Pi_H(n) := \max_{S \subset X : |S|=n} |\{H|_S\}|$, where $H|_S$ is the family of binary classifiers on the sample $S$ realized by $H$.

When $\mathscr{P}$ consists of just a single distribution, uniform convergence fails to characterize learnability. Here Benedek and Itai (1991) instead give a characterization based on finite covers.

**Definition 10 (Covers)** *For any $\varepsilon > 0$ and class $(D, X, H)$, we say $C \subseteq H$ is an $\varepsilon$-cover if every $h \in H$ is close to some $c \in C$:*

$$\forall h \in H, \exists c \in C : d_D(c, h) \leq \varepsilon$$

A class has bounded *metric entropy* if it has finite covers for every $\varepsilon$.

**Definition 11 (Metric Entropy)** *We say a class $(D, X, H)$ has metric entropy $m(\varepsilon)$ if for all $\varepsilon > 0$, $(D, X, H)$ has an $\varepsilon$-cover of size at most $m(\varepsilon)$.*

Benedek and Itai (1991) proved $(D, X, H)$ is PAC-learnable if and only if it has finite metric entropy, and conjectured a similar equivalence for the following generalization to $(\mathscr{P}, X, H)$.

**Definition 12 (Uniformly Bounded Metric Entropy)** *We say $(\mathscr{P}, X, H)$ has uniformly bounded metric entropy (UBME) if there exists $m_{UB}(\varepsilon)$ such that for all $\varepsilon > 0$ and $D \in \mathscr{P}$, $(D, X, H)$ has an $\varepsilon$-cover of size at most $m_{UB}(\varepsilon)$. We call minimum such $m_{UB}$ the metric entropy of $(\mathscr{P}, X, H)$.*

UBME is known to be equivalent to PAC-learnability in the distribution-free setting (Haussler, 1992), but is not sufficient in the general distribution-family setting (Dudley et al., 1994).

## 2.2. Density Estimation

Our work considers connections between the distribution-family model and several new variants of density estimation. We first recall the standard model adapted to our setting where the learner wishes to output an estimate to the adversary's choice of $D \in \mathscr{P}$ with respect to the $H \Delta H$-distance.

**Definition 13 (Density Estimation)** *We say $(\mathscr{P}, X, H)$ has a density estimator (DE) if there exists a function $n = n_{DE}(\varepsilon, \delta)$ such that for all $\varepsilon, \delta > 0$ there is an algorithm $A$ mapping samples $T \in X^n$ to distributions which for all $D \in \mathscr{P}$ satisfies:*

$$\Pr_{T \sim D^n}[TV_{H \Delta H}(A(T), D) > \varepsilon] \leq \delta.$$

In the context of PAC-learning, standard density estimation (even restricted to $H \Delta H$) is needlessly strong: the learner really need only estimate the distances for *most $h \in H$*, provided it *knows* the set of well-estimated events. We call this condition *intermediate density estimation*.

**Definition 14 (Intermediate Density Estimation)** *A class $(\mathscr{P}, X, H)$ has an intermediate density estimator if there exists a function $n = n_{IDE}(\varepsilon, \delta)$ such that for all $\varepsilon, \delta > 0$ there is an algorithm mapping samples $T \in X^n$ to pairs consisting of an estimator $\tilde{d}_T : H \times H \to \mathbb{R}^+$ and a subset $G_T \subseteq H$ which for all $D \in \mathscr{P}$ approximates $d_D$ in the following sense:*

1. *$\tilde{d}_T$ accurately estimates $d_D$ on $G_T$:*

$$\Pr_{T \sim D^n}[\exists g, g' \in G_T : |\tilde{d}_T(g, g') - d_D(g, g')| > \varepsilon] < \delta,$$

2. *Any fixed hypothesis occurs in $G_T$ with high probability:*

$$\forall h \in H : \Pr_{T \sim D^n}[h \in G_T] \geq 1 - \delta.$$

*We will occasionally consider specific values of $\varepsilon, \delta$, for which we write $(\varepsilon, \delta)$-IDE.*

While a weakening of the standard model, requiring knowledge of well-estimated events is still a strong assumption (and one we exploit in Section 3.1 to construct learnable classes failing IDE). With this in mind, we introduce *weak density estimation* (WDE) which drops this constraint.

**Definition 15 (Weak Density Estimation)** *A class $(\mathscr{P}, X, H)$ has a weak density estimator if there exists a function $n = n_{WDE}(\varepsilon, \delta)$ such that for all $\varepsilon, \delta > 0$ there exists an algorithm mapping samples $T \in X^n$ to estimators $\tilde{d}_T : H \times H \to \mathbb{R}^+$ which for all $D \in \mathscr{P}$ approximates $d_D$ in the following sense:*

1. *There exists a subset $G_{T,D} \subseteq H$ such that $\tilde{d}_T$ accurately estimates $d_D$ on $G_{T,D}$:*

$$\Pr_{T \sim D^n}[\exists g, g' \in G_{T,D} : |\tilde{d}_T(g, g') - d_D(g, g')| > \varepsilon] < \delta,$$

2. *Any fixed hypothesis occurs in $G_{T,D}$ with high probability:*

$$\forall h \in H : \Pr_{T \sim D^n}[h \in G_{T,D}] \geq 1 - \delta.$$

*We will occasionally consider specific values of $\varepsilon, \delta$, for which we write $(\varepsilon, \delta)$-WDE.*

In other words, the only difference between intermediate and weak density estimation is whether the set $G$ of 'good' elements can depend on the underlying distribution, or equivalently, whether $G$ is 'known' by the learner (which sees the sample but not the distribution).

## 2.3. Further Related Work

**Distribution Family Model.** The distribution-family model is implicit in the seminal works of Vapnik and Chervonenkis (1971, 1974), who studied general models of learning under convergence of empirical means. These bounds were later sharpened by Natarajan (1992). In 1991, Benedek and Itai (1991) explicitly introduced the model and conjectured its characterization by UBME which was falsified by Dudley et al. (1994) several years later. In 1997 Kulkarni and Vidyasagar (1997) observed the sufficiency of UBME in the finite and distribution-free settings could be slightly generalized to finitely coverable families and families with an interior point. Recently,[3] Lechner et al.

---

3. A version of this work appeared publicly before Lechner et al. (2023) and has been updated to reflect their result.

(2023) proved the distribution-family model has no scale-insensitive characterizing dimension.

**Density Estimation.** Density estimation (also called 'improper' distribution learning (Diakoniko-las, 2016)) is among the most classical problems in statistics. We refer the reader to Devroye and Lugosi (2001) for the most relevant overview of the area. Our techniques draw from combinatorial methods in density estimation due to Yatracos (1985) further developed in Devroye and Lugosi (2001) and the distribution learning literature (Chan et al., 2014; Diakonikolas, 2016; Ashtiani et al., 2018). Several of our results leverage ideas from this line, in particular Yatracos' method of lifting empirical estimates on covers to full density estimates.

The relation between variants of density estimation and supervised learning is also studied in Ben-David and Ben-David (2011) under the 'Known-Labeling Classifier Learning' (KLCL) model. This setting, which looks at *distribution-free* learning when a (non-realizable) labeling is known to the learner, is incomparable to our own and requires a different set of techniques.

**Unsupervised learning.** Our work also takes inspiration from unsupervised techniques in machine learning, especially multi-stage classifiers that infer information from unlabeled data before applying supervised methods (Balcan and Blum, 2010; Beimel et al., 2013; Alon et al., 2019; Hopkins et al., 2020c; Diakonikolas et al., 2022; Blanc et al., 2023), but also by similar techniques from transfer learning (Pan and Yang, 2009), active learning (Balcan et al., 2007; Hanneke and Yang, 2015; Kane et al., 2017; Hopkins et al., 2020a,b,c), privacy (Beimel et al., 2013; Alon et al., 2019), and even distribution-free learning of halfspaces (Diakonikolas et al., 2021, 2022). Most recently, a similar approach was used by Hopkins et al. (2024) to give generic reductions from agnostic to realizable settings across many variants of supervised learning (including the distribution-family model). Our notion of weak density estimation draws inspiration from their work.

**Learning Beyond Uniform Convergence.** Our work also fits into a long line of literature studying learnability beyond uniform convergence, either through notions of *local* complexity (e.g. Panchenko (2002); Bartlett et al. (2005)) or models where uniform convergence and empirical risk minimization fail completely (e.g. Shalev-Shwartz et al. (2010); David et al. (2016)). The former works relate qualitatively to our weakened notions of estimation which only require good estimates for a subset of hypotheses in the class. Most relevant, however, is work in the latter line of Shalev-Shwartz et al. (2010), who characterize Vapnik's General Learning model (which fails uniform convergence and has no combinatorial characterization (Ben-David et al., 2019)) via stability. Their work differs from our setting both in that the model is distribution-free, and that stability is a property of a learning rule while we study direct connections between the class and distribution family themselves.

## 3. Results and Proof Overview

We now overview our main results and sketch their proofs. In Section 3.1 we show IDE is sufficient but not necessary for learning. In Section 3.2 we show WDE is necessary but not sufficient. In Section 3.3 we overview the connections between WDE, UBME, and Uniform Estimation.

### 3.1. PAC Learning vs. Intermediate Density Estimation

In this section we study the relation between PAC Learning and intermediate density estimation. All proofs are sketched or otherwise shortened, and detailed versions can be found in Section 4. We start with the simple observation that (under UBME), IDE is sufficient to PAC-learn.

**Proposition 16** *Let $(\mathscr{P}, X, H)$ be any class with metric entropy $m(\varepsilon)$ and an intermediate density estimator on $n_{IDE}(\varepsilon, \delta)$ samples. Then $(\mathscr{P}, X, H)$ has a PAC learner satisfying*

$$n_{PAC}(\varepsilon, \delta) \leq O\left(n_{IDE}(\varepsilon/4, \delta/4) + \frac{\log(m(\varepsilon/4)) + \log(1/\delta)}{\varepsilon}\right).$$

**Proof** Run the IDE on an unlabeled sample $T$ of size $n_{\mathrm{IDE}}(\varepsilon/4, \delta/4)$ to get the approximate metric $\tilde{d}_T$ and set of well-estimated functions $G_T$. We are promised that with probability $1 - \delta/4$:

$$\forall g, g' \in G_T : |\tilde{d}_T(g, g') - d_D(g, g')| \leq \varepsilon/4.$$

Assume this is the case. Because $G_T$ is known, we can find $D' \in \mathscr{P}$ which realizes these estimates on $G_T$ up to an error of $\varepsilon/4$, and therefore satisfies $TV_{G_T \Delta G_T}(D, D') \leq \varepsilon/2$. Since $(\mathscr{P}, H)$ has UBME, there exists an $\frac{\varepsilon}{4}$-cover $C_T$ of $G_T$ under $D'$ of size at most $m(\varepsilon/4)$. It is enough to argue that with probability at least $1 - \delta/2$, there exists a hypothesis in $C_T$ with true risk at most $\frac{3\varepsilon}{4}$. In this case, if we simply draw $O(\frac{\log|C_T| + \log(1/\delta)}{\varepsilon})$ fresh labeled samples and output the hypothesis with lowest empirical error, a Chernoff and Union bound imply the resulting hypothesis has error at most $\varepsilon$ with probability at least $1 - \delta/4$. By a union bound, all steps in this process succeed simultaneously except with probability $\delta$, which gives the desired learner.

It is left to show a good hypothesis exists in $C_T$ with high probability. To see this, observe that if $TV_{G_T \Delta G_T}(D, D') \leq \varepsilon/2$ (which occurs except with probability $\delta/4$), $C_T$ is a $3\varepsilon/4$-cover for $G_T$ under the true distribution $D$. This follows simply from noting that for any $g, g' \in G_T$ we have:

$$d_D(g, g') = d_D(g, g') - d_{D'}(g, g') + d_{D'}(g, g') \leq TV_{G_T \Delta G_T}(D, D') + d_{D'}(g, g'),$$

so distances in the cover under $D$ change by at most $\varepsilon/2$ from their value under $D'$. On the other hand, we are promised by the IDE that $h \in G_T$ with probability at least $1 - \delta/4$. If both events occur, $h$ is covered by $C_T$, meaning $\exists h' \in C_T$ such that $d_D(h, h') \leq 3\varepsilon/4$ as desired. ∎

We note this generalizes the result of Kulkarni and Vidyasagar (1997) that UBME is sufficient for PAC learning under finitely coverable distribution families as these have simple density estimators. We now show the more involved direction: intermediate density estimation fails necessity.

**Theorem 17** *There exists a class $(\mathscr{P}, X, H)$ which is PAC-learnable in $n_{PAC}(\varepsilon, \delta) \leq \tilde{O}\left(\frac{\log(1/\delta)}{\varepsilon^2}\right)$ samples, but has no intermediate density estimator.*

We overview the construction, which is based on dictators on the noisy cube. Let $X_n = \{0, 1\}^n$ denote the $n$-dimensional hypercube, and for $x \in X_n$ write $x_i$ to denote the $i$th coordinate of $x$. Our hypothesis class $H_n := \{\mathbf{1}_i : i \in [n]\}$ will consist of the set of dictators on the cube:

$$\mathbf{1}_i(x) = \begin{cases} 1 & \text{if } x_i = 1 \\ 0 & \text{else.} \end{cases}$$

For every $x \in \{0, 1\}^n$ and $\rho > 0$, let $D_x^\rho$ denote the distribution that samples $y \in \{0, 1\}^n$ by independently setting each bit $y_i$ to $x_i$ with probability $1 - \rho$, and otherwise to $1 - x_i$. Let $\mathscr{P}_n^\rho := \{D_x^\rho\}_{x \in X_n}$ be the family of all such distributions. We argue $(\mathscr{P}_n^\rho, X_n, H_n)$ is learnable up to error $\rho$ in $O_\delta(1)$ samples, while any IDE requires $\Omega_\rho(\log(n))$ samples. The result then follows from taking an infinite disjoint union with $n \to \infty$ and $\rho \to 0$. We focus here on fixed $n, \rho$ and give the full construction in Section 4. We'll start with learnability, which simply stems from the fact that our class is 'almost constant,' and can therefore be easily approximated by a constant function.

**Lemma 18** *For any $\frac{1}{6} \geq \rho \geq 0$, $(\mathscr{P}_n^\rho, X_n, H_n)$ is $(\rho, \delta)$-PAC-learnable in $O(\log(1/\delta))$ samples.*

**Proof** Observe that by construction, no matter the distribution and hypothesis chosen by the adversary our class is almost constant in the following sense:

$$\forall D_x^\rho \in \mathscr{P}_n^\rho, h \in H_n, \exists z \in \{0, 1\} : \Pr_{y \sim D_x^\rho}[h(y) = z] = 1 - \rho.$$

This suggests the following strategy: after $O(\log \frac{1}{\delta})$ samples, output the majority label as a constant function.[4] For $\rho \leq 1/6$, Chernoff promises the learner picks the true majority with probability at least $1 - \delta$. Since the majority label has mass $1 - \rho$, this gives the desired $(\rho, \delta)$-PAC-learner. ∎

Density estimation is somewhat more involved. At a high level, our lower bound stems from the fact that the intermediate estimator must 'declare' a set of good elements, but the independent noise across coordinates inherent in $\mathscr{P}_n^\rho$ makes this challenging without a large number of samples.

**Proposition 19** *For all $n > 2$ and $\frac{1}{6} > \rho > \frac{12 \log(1/\rho)}{\log(n)}$, any IDE for $(\mathscr{P}_n^\rho, X_n, H_n)$ satisfies*

$$n_{IDE}(1/6, 1/8) \geq \Omega\left(\frac{\log(n)}{\log(1/\rho)}\right).$$

Formally, it is convenient to reduce Proposition 19 to showing no algorithm can 'confidently' compute even a portion of the string $x$ underlying some $D_x^\rho \in \mathscr{P}_n^\rho$ in the following sense.

**Lemma 20** *For any $t \in \mathbb{N}$ and $\frac{1}{6} > \rho > 0$, if $(\mathscr{P}_n^\rho, X_n, H_n)$ has a $(1/6, 1/8)$-IDE on $t$ samples, then there exists a (randomized) algorithm $\mathcal{A} : X^t \to \{0, 1, \bot\}$ with the following guarantee. Given $t$ samples drawn from any $D_x^\rho \in \mathscr{P}_n^\rho$, with probability at least $1/8$, $\mathcal{A}$ outputs a string $\hat{x} \in \{0, 1, \bot\}^n$ such that:*

*1. $\hat{x}$ is '$\bot$' on at most $n/2$ coordinates,*

*2. $\hat{x}$ agrees with $x$ otherwise.*

*We call any such $\mathcal{A}$ a $(1/2, 1/8)$-confident learner for $\mathscr{P}_n^\rho$.*

The proof, which follows from observing that $\tilde{d}_T$ can be used to compute $x_i$ whenever $\mathbf{1}_i \in G_T$, is deferred to Section 4 (Lemma 33). It is now enough to show no such learner exists.

**Proof** [Sketch, Proposition 19] By Lemma 20 and Yao's minimax principle, it is enough to exhibit a distribution over $\mathscr{P}_n^\rho$ such that any deterministic confident learner on $t$ samples fails with probability

---

4. We note this can be made 'proper' simply by adding the all 0's and all 1's function to the class. This only makes IDE harder, so it does not effect the following lower bound.

at least $7/8$. We will use the uniform distribution over $\mathscr{P}_n^\rho$, which corresponds to choosing $D_x^\rho$ for a uniformly random $x \in \{0,1\}^n$. Observe it is sufficient to consider learners $\mathcal{A}$ that output $\bot$ on the noisiest $\frac{n}{2}$ coordinates[5] and output the majority bit otherwise, since these maximize the learner's success rate (see Proposition 32 for details). The idea is to then observe that unless $t$ is sufficiently large, for any choice of $D_x^\rho$ the following two events almost certainly occur across $T \sim (D_x^\rho)^t$:

1. There is a coordinate $i \in [n]$ flipped by noise in every example: $\forall y \in T : y_i \neq x_i$

2. There are at least $n/2$ coordinates $i \in [n]$ with empirical density $0 < \frac{1}{t} \sum_{y \in T} y_i < 1$.

$\mathcal{A}$ fails on any such input by construction, since it outputs the empirical majority on a flipped coordinate. The probability that there exists at least one coordinate that is flipped in each of $t$ samples is $1 - (1 - \rho^t)^n \geq 15/16$ by our choice of constants. On the other hand, the expected number of coordinates with empirical density $0$ or $1$ is at most $2n(1 - \rho)^t \leq 2ne^{-\rho t} \leq \frac{n}{32}$ by our assumptions on $\rho$. By Markov's inequality and a union bound both conditions hold with probability greater than $7/8$, so the algorithm cannot succeed with more than $1/8$ probability as desired. ∎

## 3.2. PAC Learning vs. Weak Density Estimation

We now show weak density estimation is necessary but not sufficient to PAC-learn. All proofs are sketched or otherwise shortened, and detailed versions can be found in Section 5.

**Proposition 21** *Let $(\mathscr{P}, X, H)$ be a class that is PAC-learnable in $n_{PAC}(\varepsilon, \delta)$ samples. Then $(\mathscr{P}, X, H)$ has a weak density estimator satisfying*

$$n_{WDE}(\varepsilon, \delta) \leq O\left(n_{PAC}(\varepsilon/3, \delta) + \frac{\log(\Pi_H(n_{PAC}(\varepsilon/3, \delta))) + \log(1/\delta)}{\varepsilon^2}\right),$$

*where we recall $\Pi_H$ is the growth function $\Pi_H(n) := \max_{S \subset X : |S| = n} |\{H|_S\}|$.*

**Proof** Run a PAC-learner $\mathcal{L}$ across all labelings of a sample $T$ of size $n_{PAC}(\varepsilon/3, \delta)$ to generate

$$C_T := \{\mathcal{L}(T, h(T)) : h \in H\}.$$

Hopkins et al. (2024) observed $C_T$ is a 'non-uniform cover' in the sense that for each $h \in H$, if we define $c_T : H \to C_T$ by $c_T(h) := \mathcal{L}(T, h(T))$, then $c_T(h)$ is close to $h$ with high probability:

$$\forall h \in H : \Pr_T[d_D(h, c_T(h)) > \varepsilon/3] < \delta.$$

To build our WDE, we directly estimate distances on $C_T$ and extend to $H$ via $c_T$. Draw a fresh sample $T'$ of $O(\frac{\log|C_T| + \log(1/\delta)}{\varepsilon^2})$ unlabeled examples. By Chernoff and Union bounds, the empirical estimator $\hat{d}_{T'}(h, h') := \frac{1}{|T'|} \sum_{x \in T'} \mathbf{1}\{h(x) \neq h'(x)\}$ is close to $d_D$ with high probability on $C_T$:

$$\Pr_{T'}\left[\exists h, h' \in C_T : |\hat{d}_{T'}(h, h') - d_D(h, h')| > \varepsilon/3\right] < \delta.$$

---

5. I.e. coordinates with empirical density closest to $1/2$, ties broken arbitrarily.

We can extend $\hat{d}_{T'}$ to an approximate estimator over all of $H$ by defining for all $h, h' \in H$:

$$\tilde{d}_{T \cup T'}(h, h') := \hat{d}_{T'}(c_T(h), c_T(h')).$$

We argue this estimate is accurate across all hypotheses which are close to their image under $c_T$:

$$G_{T \cup T', D} := \{g \in H : d_D(g, c_T(g)) \le \varepsilon/3\}.$$

Assuming our empirical estimates on $C_T$ are good, note for any $g, g' \in G_{T \cup T', D}$ we can write:

$$
\begin{aligned}
|\tilde{d}_{T \cup T'}(g, g') - d_D(g, g')| &= |\hat{d}_{T'}(c_T(g), c_T(g')) - d_D(g, g')| \\
&\le |\hat{d}_{T'}(c_T(g), c_T(g')) - d_D(c_T(g), c_T(g'))| + d_D(g, c_T(g)) + d_D(g', c_T(g')) \\
&\le \varepsilon,
\end{aligned}
$$

Thus we have that each $h \in H$ lies in $G_{T \cup T', D}$ with probability at least $1 - \delta$, and all estimates in $G_{T \cup T', D}$ are $\varepsilon$-accurate with probability at least $1 - \delta$, which gives the desired WDE. ∎

Given the closeness of weak and intermediate density estimation, one might hope to amplify weak estimation to a PAC-learner. Unfortunately, the main barrier to such a proof (lacking knowledge of the well-estimated set) is inherent, and we are able to give a very strong refutation of WDE's sufficiency: even a *perfect* weak estimator (i.e. one with error $\varepsilon = 0$) isn't sufficient to PAC-learn.

**Theorem 22** *There exists a class $(\mathscr{P}, X, H)$ with metric entropy $m(\varepsilon) \le O(2^{1/\varepsilon})$ and a WDE on $n_{WDE}(0, \delta) \le O(\log(1/\delta))$ samples which is not PAC-learnable.*

We first describe the construction, which is similar in spirit to the UBME counterexample of Dudley et al. (1994). Let $X = \{0, 1, 2\}^{\mathbb{N}}$. For each $i \in \mathbb{N}$, let $\varepsilon_i := \frac{1}{\log(i+255)}$, define $f_i := \mathbf{1}[x_i = 0]$, and $D_i$ to be the distribution over $x \in X$ where $x_i$ is chosen from $\{0, 1\}$ uniformly at random and all remaining coordinates $j \neq i$ are drawn independently from the categorical distribution

$$\Pr[x_j = 0] = \Pr[x_j = 1] = \epsilon_j, \text{ and } \Pr[x_j = 2] = 1 - 2\epsilon_j.$$

Finally, define $\mathscr{P} = \{D_i\}_{i \in \mathbb{N}}$ and $H = \{f_i\}_{i \in \mathbb{N}}$. We first argue that $(\mathscr{P}, X, H)$ has a WDE.

**Proposition 23** *$(\mathscr{P}, X, H)$ has a weak density estimator on $n_{WDE}(0, \delta) \le O(\log(1/\delta))$ samples.*

**Proof** Fix any $f_j, f_k$, and observe that across the choices of $D_i$, $d_{D_i}(f_j, f_k)$ depends only on whether $i \in \{j, k\}$. Namely, for any $j \neq i$ we have $d_{D_i}(f_i, f_j) = \frac{1}{2}$, while any $i \neq j \neq k$ satisfy $d_{D_i}(f_j, f_k) = \epsilon_j(1 - \epsilon_k) + \epsilon_k(1 - \epsilon_j)$. Thus we can compute distances simply by checking if $i \in \{j, k\}$. With this in mind, given a sample $T$ of size $O(\log(1/\delta))$, let $\mu_\ell^T := \frac{1}{t} \sum_{y \in T} y_\ell$ and define:

$$
\tilde{d}_T(f_j, f_k) = \begin{cases} \epsilon_j(1 - \epsilon_k) + \epsilon_k(1 - \epsilon_j) & \text{if } \mu_j^T, \mu_k^T < 1/3 \\ 1/2 & \text{otherwise,} \end{cases}
$$

and let $G_{T, D}$ be the set of functions with good empirical averages:

$$G_{T, D} := \left\{ f_i : \left| \underset{D}{\mathbb{E}}[f_i] - \mu_i^T \right| < 1/6 \right\}.$$

Fix any $D_i \in \mathscr{P}$ and $f_j \in H$. First, observe that by a Chernoff bound, $f_j \in G_{T,D_i}$ with probability at least $1 - \delta$. Second, notice that for all functions $f_\ell \in G_{T,D_i}$, we are assured that $\mu_\ell^T < 1/3$ if and only if $\ell \neq i$ (since $\mathbb{E}[f_i] = 1/2$, while $\mathbb{E}[f_\ell] \leq 1/8$ for $\ell \neq i$). Since this condition exactly determines the distances between hypotheses, $\hat{d}_T$ has zero error across $G_{T,D_i}$ as desired. ∎

Next, we claim $(\mathscr{P}, X, H)$ has small metric entropy, which simply follows from observing that for any fixed $D_i$, most hypotheses are close to the all 0's function (see Section 5, Proposition 37).

**Proposition 24** $(\mathscr{P}, X, H)$ *has metric entropy at most* $m(\varepsilon) \leq O(2^{1/\varepsilon})$.

Finally, we argue our class is not PAC-learnable. The proof has a similar flavor to our lower bound against IDEs: the learner cannot distinguish between ground truth and hypotheses that happen to look consistent due to a small amount of noise (here generated by the $\epsilon_i$'s).

**Proposition 25** $(\mathscr{P}, X, H)$ *is not PAC-learnable.*

**Proof** [Sketch] By Yao's principle it suffices to find for every $t \in \mathbb{N}$ a randomized strategy for the adversary over which any deterministic learner on $t$ samples fails with at least constant probability. For each $n \in \mathbb{N}$, consider the strategy that selects from the set $\{D_i, f_i\}_{i \in [n]}$ with probability $\Pr[(D_i, f_i)] \propto \epsilon_i^t$. For each $j \in [n]$, call a sample $S$ *consistent* with $(D_j, f_j)$ if for all $(x, y) \in S$, we have $x \in \mathrm{supp}(D_j)$ and $f_j(x) = y$. The idea is to argue that when the learner's sample is consistent with multiple coordinates, the posterior choices over consistent $(D_j, f_j)$ are uniform and the learner cannot distinguish the ground truth. On the other hand, for large enough $n \gg t$, there are almost always multiple consistent coordinates. See Section 5 (Proposition 38) for further details. ∎

## 3.3. Density Estimation, Uniform Convergence, and UBME

Finally, we compare density estimation with prior paradigms in the literature. All proofs are sketched or otherwise shortened, and detailed versions can be found in Section 6. We start by introducing a relaxed variant of uniform convergence called uniform estimation.

**Definition 26 (Uniform Estimation)** *We say* $(\mathscr{P}, X, H)$ *satisfies the uniform estimation property if there exist a family of estimators* $\{\mathcal{E}_S\}_{S \in (X \times \{0,1\})^*} : H \to \mathbb{R}_+$ *and a function* $n = n_{UE}(\varepsilon, \delta)$ *such that for all* $\varepsilon, \delta > 0$, $D \in \mathscr{P}$, *and* $h \in H$:

$$\Pr_{S \sim (D \times h)^n}[\exists h' \in H : |\mathcal{E}_S(h') - err_{D \times h}(h')| > \varepsilon] \leq \delta.$$

Uniform estimation is the natural supervised analog of density estimation, estimating $err_{D \times h}(h') = d_D(h, h')$ from *labeled* samples. We formalize this connection by showing the two are equivalent so long as the underlying class has UBME.[6]

**Theorem 27** *If* $(\mathscr{P}, X, H)$ *has a uniform estimator on* $n_{UE}(\varepsilon, \delta)$ *samples, then* $(\mathscr{P}, X, H)$ *has a density estimator satisfying*

$$n_{DE}(\varepsilon, \delta) \leq O\left(n_{UE}(\varepsilon', \delta') + \frac{\log(\Pi_H(n_{UE}(\varepsilon', \delta'))) + \log(1/\delta)}{\varepsilon^2}\right)$$

---

6. Note unlike the distribution-free case this is trivially false otherwise since uniform estimation implies UBME, but any triple $(D, X, H)$ has a trivial density estimator.

*where $\varepsilon' = O(\varepsilon)$, $\delta' = O(\frac{\delta}{\Pi_H(n_{UE}(\varepsilon/4, \delta/2))})$. Conversely, if $(\mathscr{P}, X, H)$ has a density estimator on $n_{DE}(\varepsilon, \delta)$ samples and has metric entropy $m(\varepsilon)$, then $(\mathscr{P}, X, H)$ has a uniform estimator satisfying*

$$n_{UE}(\varepsilon, \delta) \leq O\left(n_{DE}(\varepsilon/4, \delta/2) + \frac{\log(m(\varepsilon/4)) + \log(1/\delta)}{\varepsilon^2}\right).$$

Theorem 27 implicitly revolves around the fact that under UBME, both uniform and density estimation are equivalent to building a (bounded) uniform $\varepsilon$-cover $C$ and a corresponding *covering map* $c : H \to C$ such that $\forall h \in H : d_D(c(h), h) \leq \varepsilon$. This type of structure allows us to apply Yatracos (1985)'s trick extending empirical estimates on $C$ to good estimates on all of $H$. We now overview the constructions underlying Theorem 27, leaving proof details to Section 6 (Theorem 39).

**DE + UBME $\implies$ UE:** Running the UE outputs a distribution $D'$ s.t. $TV_{H \triangle H}(D', D)$ is small. UBME promises that $D'$ has a known cover $C$ and covering map $c$, which is in turn a covering map for $D$ as well (as in Proposition 16). We can then define our estimators by drawing a labeled sample $S$, and extending the empirical error of $S$ on $C$ to all of $H$ via $\mathcal{E}_S(h) := \text{err}_S(c(h))$.

**UE $\implies$ DE:** Hopkins et al. (2024) showed it is possible to build a cover $C$ for any learnable class from unlabeled samples (see Section 6, Lemma 40). We show that using the uniform estimator, it is also possible to construct $C$'s associated covering map.[7] Draw an unlabeled sample $T$, and run the uniform estimator across every possible labeling of $T$ by $C$ to obtain $\{\mathcal{E}_{(T, h(T))}\}_{h \in C}$. We define a covering map by sending $h \in H$ to its closest element in $C$ according to these estimates: $c(h) := \text{argmin}_{h' \in C} \mathcal{E}_{(T, h'(T))}(h)$. Now draw an unlabeled sample $T$, define $\tilde{d}_T$ by extending empirical estimates on $C$ to $H$ as $\tilde{d}_{T'}(h, h') := \hat{d}_{T'}(c(h), c(h'))$, and output any distribution $D' \in \mathscr{P}$ approximately satisfying all estimates.

It is left to examine the connection between density estimation and UBME. We show the latter actually directly implies weak density estimation.

**Proposition 28** *Any class $(\mathscr{P}, X, H)$ with metric entropy $m(\varepsilon)$ has a WDE satisfying*

$$n_{WTV}(\varepsilon, \delta) \leq O\left(\frac{\log(m(\varepsilon/8)) + \log(1/\delta)}{\varepsilon^2}\right).$$

**Proof** [Sketch] We argue it is sufficient to simply output the empirical distance estimator on a large enough unlabeled sample $T$. UBME promises the existence of an (unknown) cover $C$ and covering map $c$. Let $G_{T, D}$ be the set of all $h \in H$ s.t. $d_D(h, c(h))$ is estimated to within $O(\varepsilon)$ error. The idea is to observe that the empirical distance of any two $h, h' \in G_{T, D}$ can be factorized through their distances to $c(h)$ and $c(h')$ *without any knowledge of $c$*, since:

$$|\hat{d}_T(h, h') - d_D(h, h')| \leq |\hat{d}_T(c(h), c(h')) - d_D(c(h), c(h'))| + O(\varepsilon).$$

Thus as long as we draw a sufficient number of samples to estimate distances on $C$, just the existence of this cover is sufficient to imply $\hat{d}_T$ is a WDE. See Section 6 (Proposition 43) for details. ∎

Note that while Proposition 28 subsumes Proposition 21 since any PAC-learnable class has UBME (Benedek and Itai, 1991), they give quantitatively different sample complexity guarantees. Combined with Dudley et al. (1994), Proposition 28 also implies insufficiency of WDEs, but the approach is inherently lossy and cannot handle the 'perfect' $\varepsilon = 0$ case Theorem 22 covers.

---

7. Note this is not in general possible for any learnable class, as this would refute Theorem 17.

## Acknowledgements

SL was supported by a Simons Investigator Award (#929894). DK was supported by by NSF Medium Award CCF-2107547 and NSF Award CCF-1553288 (CAREER).

## 4. Detailed proofs of Section 3.1

In this section we study the relation between PAC Learning and intermediate density estimation. We start with the basic observation that (under UBME), intermediate estimation is sufficient to PAC-learn.

**Proposition 29** *Let $(\mathscr{P}, X, H)$ be any class with metric entropy $m(\varepsilon)$ and an intermediate density estimator on $n_{IDE}(\varepsilon, \delta)$ samples. Then $(\mathscr{P}, X, H)$ is PAC-learnable in*

$$n_{PAC}(\varepsilon, \delta) \leq O\left(n_{IDE}(\varepsilon/4, \delta/4) + \frac{\log(m(\varepsilon/4)) + \log(1/\delta)}{\varepsilon}\right)$$

*samples.*

**Proof** Fix $D \in \mathscr{P}$ and $h \in H$. At a high level, we will use the IDE to find a small subset $C \subset H$ that almost certainly contains some $h'$ close to $h$, then learn $C$ directly via empirical risk minimization.

More formally, run the IDE on an unlabeled sample $T$ of size $n_{\text{IDE}}(\varepsilon/4, \delta/4)$ to get the approximate metric $\tilde{d}_T$ and set of well-estimated functions $G_T$. With probability at least $1 - \delta/4$, all distances within $G_T$ are well-estimated:

$$\forall g, g' \in G_T : |\tilde{d}_T(g, g') - d_D(g, g')| \leq \varepsilon/4.$$

Assume this is the case. Because $G_T$ is known, we can find a distribution $D' \in \mathscr{P}$ which realizes these estimates on $G_T$ up to an error of $\varepsilon/4$, and therefore satisfies $TV_{G_T \Delta G_T}(D, D') \leq \varepsilon/2$. Since $G_T \subset H$ and $(\mathscr{P}, H)$ has UBME, there exists an $\frac{\varepsilon}{4}$-cover $C_T$ of $G_T$ under $D'$ of size at most $m(\varepsilon/4)$. It is enough to argue that with probability at least $1 - \delta/2$, there exists a hypothesis in $C_T$ with small true risk:

$$\exists h' \in C_T : \text{err}_{D \times h}(h') \leq \frac{3\varepsilon}{4}.$$

In this case, if we simply draw $O(\frac{\log |C_T| + \log(1/\delta)}{\varepsilon})$ fresh labeled samples and output the hypothesis with lowest empirical error, a Chernoff and Union bound imply the resulting hypothesis has error at most $\varepsilon$ with probability at least $1 - \delta/4$. By a union bound, all steps in this process succeed simultaneously except with probability $\delta$, which gives the desired learner.

It is left to show a good hypothesis exists in $C_T$ with high probability. To see this, observe that if $TV_{G_T \Delta G_T}(D, D') \leq \varepsilon/2$ (which we argued occurs except with probability $\delta/4$), $C_T$ is a $3\varepsilon/4$-cover for $G_T$ under the true distribution $D$. This follows simply from noting that for any $g, g' \in G_T$ we have:

$$\begin{aligned} d_D(g, g') &= d_D(g, g') - d_{D'}(g, g') + d_{D'}(g, g') \\ &\leq TV_{G_T \Delta G_T}(D, D') + d_{D'}(g, g'), \end{aligned}$$

so distances in the cover under $D$ change by at most $\varepsilon/2$ from their value under $D'$. On the other hand, we are promised by the IDE that $h \in G_T$ with probability at least $1 - \delta/4$. If both events

occur, then $h$ is covered by $C_T$, meaning there exists $h' \in C_T$ such that $d_D(h, h') \leq 3\varepsilon/4$ as desired. ∎

We note that this observation generalizes the result of Kulkarni and Vidyasagar (1997) that UBME is sufficient for PAC learning under finitely coverable distribution families as these have simple density estimators. With this out of the way we move to the more involved direction, showing intermediate density estimation fails necessity.

**Theorem 30** *There exists a class $(\mathscr{P}, X, H)$ which is PAC-learnable in*

$$n_{PAC}(\varepsilon, \delta) \leq \tilde{O}\left(\frac{\log(1/\delta)}{\varepsilon^2}\right)$$

*samples, but has no intermediate density estimator.*

We first overview the construction, which is based on learning dictators on the noisy hypercube. Let $X_n = \{0,1\}^n$ denote the $n$-dimensional hypercube, and for $x \in X_n$ write $x_i$ to denote the value of the $i$th coordinate of $x$. Our hypothesis class $H_n := \{\mathbf{1}_i : i \in [n]\}$ will consist of the set of dictators on the cube:

$$\mathbf{1}_i(x) = \begin{cases} 1 & \text{if } x_i = 1 \\ 0 & \text{else.} \end{cases}$$

Finally, for every $x \in \{0,1\}^n$ and $\rho > 0$, let $D_x^\rho$ denote the distribution that samples $y \in \{0,1\}^n$ by independently setting each bit $y_i$ to $x_i$ with probability $1 - \rho$, and otherwise to $1 - x_i$. We equivalently think of $y$ as being generated by applying independent noise to each bit of $x$, hence the relation to the noisy cube.

Let $\mathscr{P}_n^\rho := \{D_x^\rho\}_{x \in X_n}$ be the family of all such distributions. We argue that $(\mathscr{P}_n^\rho, X_n, H_n)$ is learnable up to error $\rho$ in $O_\delta(1)$ samples, while any IDE requires $\Omega_\rho(\log(n))$ samples. The result then follows from taking an appropriate infinite disjoint union with $n \to \infty$ and $\rho \to 0$.

We start with the easier of the two, learnability, which simply stems from the fact that our class is 'almost constant,' and can therefore be easily approximated by a constant function.

**Lemma 31** *For any $\frac{1}{6} \geq \rho \geq 0$, $(\mathscr{P}_n^\rho, X_n, H_n)$ is $(\rho, \delta)$-PAC-learnable in $O(\log(1/\delta))$ samples.*

**Proof** Observe that by construction, no matter the distribution and hypothesis chosen by the adversary our class is almost constant in the following sense:

$$\forall D_x^\rho \in \mathscr{P}_n^\rho, h \in H_n, \exists z \in \{0,1\} : \Pr_{y \sim D_x^\rho}[h(y) = z] = 1 - \rho.$$

This follows from the fact that for all $x \in \{0,1\}^n$ and $i \in [n]$:

$$\Pr_{y \sim D_x^\rho}[y_i = x_i] = 1 - \rho,$$

and every fixed $h \in H_n$ depends only on the value of a single coordinate.

This suggests the following strategy: after $O(\log \frac{1}{\delta})$ samples, simply output the majority label as a constant function.[8] For any $\rho$ bounded away from $\frac{1}{2}$, a Chernoff bound promises that the learner

---

8. We note this can be made 'proper' simply by adding the all 0's and all 1's function to the class. This only makes density estimation harder, so it does not effect the following lower bound.

picks the true majority label with probability at least $1 - \delta$. Since the majority label has mass $1 - \rho$, this gives the desired $(\rho, \delta)$-PAC-learner. ∎

Density estimation is somewhat more involved. At a high level, our lower bound stems from the fact that the intermediate estimator must 'declare' a set of good elements, but the independent noise across coordinates inherent in $\mathscr{P}_n^\rho$ makes this challenging without a large number of samples.

**Proposition 32** *For all $n > 2$ and $\frac{1}{6} > \rho > \frac{12 \log(1/\rho)}{\log(n)}$, any IDE for $(\mathscr{P}_n^\rho, X_n, H_n)$ uses at least*

$$n_{IDE}(1/6, 1/8) \geq \Omega\left(\frac{\log(n)}{\log(1/\rho)}\right)$$

*samples.*

The crux of Proposition 32 lies in the fact that, given a few samples from a random $D_x^\rho$, while it may be easy to output some $x'$ close to $x$ with high probability it is impossible to do so *confidently*. In other words, the learner should not be able to identify a large set of coordinates on which it knows it is correct due to the inherent noise in $D_x^\rho$.[9] We formalize this connection to confident learning via the following reduction.

**Lemma 33** *For any $t \in \mathbb{N}$ and $\frac{1}{6} > \rho > 0$, if $(\mathscr{P}_n^\rho, X_n, H_n)$ has a $(1/6, 1/8)$-IDE on $t$ samples, then there exists a (randomized) algorithm $\mathcal{A} : X^t \rightarrow \{0, 1, \bot\}$ with the following guarantee. Given $t$ samples drawn from any $D_x^\rho \in \mathscr{P}_n^\rho$, with probability at least $1/8$, $\mathcal{A}$ outputs a string $\hat{x} \in \{0, 1, \bot\}^n$ such that:*

  *1. $\hat{x}$ is '$\bot$' on at most $n/2$ coordinates,*

  *2. $\hat{x}$ agrees with $x$ otherwise.*

*We call any such $\mathcal{A}$ a $(1/2, 1/8)$-confident learner for $\mathscr{P}_n^\rho$.*

**Proof** Consider the following algorithm for generating $\hat{x}$. Draw a sample $T$ of $t$ elements from $D_x^\rho$, and run the IDE to get $\tilde{d}_T$ and $G_T$. Pick any $\mathbf{1}_i \in G_T$ (if no such hypothesis exists output the all '$\bot$' string), and randomly guess the value of its corresponding coordinate in $x$:

$$\hat{x}_i \sim \text{Ber}(1/2).$$

For every other $\mathbf{1}_j \in G_T$, define $\hat{x}_j$ by its approximate distance to $\mathbf{1}_i$:

$$\hat{x}_j = \begin{cases} \hat{x}_i & \text{if } d_{\tilde{T}}(\mathbf{1}_i, \mathbf{1}_j) < 1/2 \\ 1 - \hat{x}_i & \text{otherwise.} \end{cases}$$

Output "$\bot$" in all other circumstances.

At a high level, this generation procedure works because $\hat{x}_i$ is correct half the time, and the true value of any $x_j$ is reflected in its indicator's distance $d_{D_x^\rho}(\mathbf{1}_i, \mathbf{1}_j)$. More formally, observe that whenever $x_i = x_j$, the distance between the indicators of $i$ and $j$ is

$$d_{D_x^\rho}(\mathbf{1}_i, \mathbf{1}_j) \leq 2\rho < 1/3,$$

---

9. Algorithms with this type of guarantee are well-studied in the learning literature under a variety of names such as "Reliable and Probably Useful" (Rivest and Sloan, 1988), "Confident" (Kane et al., 2017), or "Knows What It Knows" (Li et al., 2001).

whereas whenever $x_i \neq x_j$, we have

$$d_{D_x^\rho}(\mathbf{1}_i, \mathbf{1}_j) \geq 1 - 2\rho > 2/3.$$

With probability at least $7/8$, $\tilde{d}_T$ is $1/6$-accurate on $G_T$, meaning we can distinguish between these two cases and correctly identify whether $x_i = x_j$. Then as long as $\hat{x}_i = x_i$ (which occurs with probability $1/2$), the above procedure correctly labels all indices whose indicators are in $G_T$. Thus as long as $|G_T| \geq \frac{n}{2}$, our output $\hat{x}$ satisfies the desired properties. Since every hypothesis lies in $G_T$ with probability at least $7/8$, this occurs with probability at least $3/4$ by Markov. Union bounding over all events, our algorithm succeeds with probability at least $1/8$ as desired.

■

We are now ready to prove Proposition 32 by showing $\mathscr{P}_n^\rho$ has no confident learner.

**Proof** [Proof of Proposition 32] It is enough to prove $\mathscr{P}_n^\rho$ has no $(1/2, 1/8)$-confident learner on $t = \frac{2\log(n)}{5\log(1/\rho)}$ samples. By Yao's minimax principle, it is enough to show there is a distribution over $\mathscr{P}_n^\rho$ such that any deterministic algorithm on $t$ samples fails with probability at least $7/8$. We will use the uniform distribution over $\mathscr{P}_n^\rho$, which corresponds to choosing $D_x^\rho$ for a uniformly random $x \in \{0,1\}^n$.

To simplify the proof, we first argue it is sufficient to consider learners that adhere to the following 'noisy majority rule', that is mappings $\mathcal{A}$ such that:

1. $\mathcal{A}$ outputs $\perp$ on the noisiest[10] $\frac{n}{2}$ coordinates,

2. $\mathcal{A}$ outputs the majority bit otherwise.

To see this, we argue any learning rule can be updated to satisfy the above constraints in a way that only improves its success probability. For any $x$, let $\mathcal{W}_x \subset \{0, 1, \perp\}^n$ denote the set of 'winning' strings, that is those that label at least half the coordinates and always agree with $x$. Thinking of the choice of $x$ and $T$ as a jointly distributed variable, expand the success probability of $A$ as:

$$\Pr_{x \sim \{0,1\}^n, T \sim (D_x^\rho)^t}[A(T) \in \mathcal{W}_x] = \sum_{T_0 \in X_n^t} \Pr_{x,T}[T = T_0] \Pr_{x,T}[A(T_0) \in \mathcal{W}_x | T = T_0].$$

Thus we may make any update to the algorithm's output $A(T_0)$ which cannot decrease the conditional success probability $\Pr[A(T_0) \in \mathcal{W}_x | T = T_0]$. We argue that we can always locally update the output of the algorithm to one following noisy majority without decreasing the success probability.

First, note we may assume without loss of generality that $A(T_0)$ labels (answers '1' or '0' on) exactly half the coordinates. If $A(T_0)$ labels more than half, replacing labels with $\perp$ up to $\frac{n}{2}$ can only increase the probability $A(T_0) \in \mathcal{W}_x$, while if $A(T_0)$ labels fewer than half, we automatically have $A(T_0) \notin \mathcal{W}_x$ so replacing $\perp$ even with arbitrary labels up to $n/2$ also only increases this probability. With this in mind, let $\mathcal{L}_{T_0}$ denote the set of $n/2$ labeled coordinates in $A(T_0)$. Notice that even after conditioning on $T = T_0$, the coordinates of $x$ in the posterior remain independent

---

10. I.e. coordinates with empirical density closest to $1/2$, ties broken arbitrarily.

random variables, so we can expand the conditional success probability as:

$$\Pr_{x,T}[A(T_0) \in \mathcal{W}_x | T = T_0] = \Pr_{x,T}\left[\bigwedge_{i \in \mathcal{L}_{T_0}} \{A(T_0)_i = x_i\} \middle| T = T_0\right]$$

$$= \prod_{i \in \mathcal{L}_{T_0}} \Pr_{x,T}[A(T_0)_i = x_i | T = T_0].$$

The posterior probability of any particular $x_i$ is maximized by the coordinate's majority value $\text{maj}_{y \in T_0}\{y_i\}$, so we can always set $A(T_0)_i$ to the majority value without decreasing the overall success probability.

To ensure we only label the noisiest coordinates, observe further that the posterior probability of the majority value increases the further its empirical mean $\mu_i^{T_0} = \frac{1}{t}\sum_{y \in T_0} y_i$ is from $1/2$. Since all the coordinates are identically distributed conditioned on having the same empirical mean by symmetry, any coordinate pair $(i,j)$ satisfying

1. $A(T_0)$ labels $j$ but not $i$:
$$A(T_0)_i = \bot, \quad A(T_0)_j \in \{0,1\}$$

2. Coordinate $j$ is noisier than $i$:
$$\left|\mu_j^T - \frac{1}{2}\right| < \left|\mu_i^T - \frac{1}{2}\right|$$

can be swapped to setting $A(T_0)_i$ to its majority label and $A(T_0)_j$ to $\bot$ without decreasing the success probability. Iteratively applying these updates leads to satisfying noisy majority.

It is left to show that any algorithm following the noisy majority rule fails with high probability. To see this, observe that unless $t$ is sufficiently large, for any choice of $D_x^\rho$ the following two events almost certainly occur:

1. There is a coordinate whose value is flipped by noise in every example:
$$\exists i, \forall y \in T : y_i \neq x_i$$

2. There are at least $n/2$ coordinates with empirical density $0 < \mu_i^T < 1$.

Noisy majority fails on any such input by construction, since it outputs majority label of a flipped coordinate. The probability there exists at least one coordinate that is flipped in each of $t$ samples is $1 - (1 - \rho^t)^n \geq 15/16$ by our choice of constants. On the other hand, the expected number of coordinates with empirical density 0 or 1 is at most $2n(1-\rho)^t \leq 2ne^{-\rho t} \leq \frac{n}{32}$ by our assumptions on $\rho$. By Markov's inequality and a union bound both conditions hold with probability greater than $7/8$, so the algorithm cannot succeed with more than $1/8$ probability as desired. ∎

Finally we argue that taking the disjoint union of infinitely many such instances gives the full result.

**Proof** [Proof of Theorem 30] Consider the sequence $\{\rho_n\}_{n \in \mathbb{N}}$ where $\rho_n = \Theta\left(\frac{\log\log(n)}{\log(n)}\right)$. For the correct choice of constant (and large enough $n$), note that the following two conditions hold:

1. $\rho_n$ satisfies the conditions of Proposition 32: $\frac{1}{6} > \rho > \frac{12 \log(1/\rho)}{\log(n)}$

2. $t_n \coloneqq \frac{2 \log(n)}{5 \log(1/\rho_n)}$ is increasing.

Consider the disjoint union of classes $\bigcup_{n \in \mathbb{N}} \mathcal{H}_n^{\rho_n}$. We first prove this class is PAC-learnable. Given any $\varepsilon, \delta > 0$, we consider two cases. In the first case, the adversary selects a distribution/hypothesis pair indexed by $n$ such that $\rho_n \leq \varepsilon$. In this case, PAC-learnability follows exactly the same argument as in Proposition 32. Otherwise, the adversary selects $n$ such that $\rho_n > \varepsilon$. By definition of $\rho_n$ it must therefore be the case that $n = |\mathcal{H}_n^{\rho_n}| \leq O\left(\left(\frac{1}{\varepsilon}\right)^{1/\varepsilon}\right)$, so standard arguments show that outputting any empirical risk minimizer over $O(\frac{\log \frac{1}{\delta}}{\varepsilon} + \frac{\log \frac{1}{\varepsilon}}{\varepsilon^2})$ samples is a PAC-learner. Finally note these cases can be distinguished by the learner in one sample, since each instance space is 'marked' by its corresponding index (value of $n$).

On the other hand, it is easy to see there cannot be any bounded $(1/6, 1/8)$-IDE for this class. For any fixed sample size $t$, the adversary can always choose $n$ such that $t_n > t$ (since $\{t_n\}$ is increasing), and corresponding $\rho_n$ satisfying the conditions of Proposition 32. Since $\mathcal{H}_n^{\rho_n}$ then has no $(1/6, 1/8)$-IED on $t$ samples, this completes the proof. ∎

We note it is possible to improve the sample complexity of the learner to $\tilde{O}(\frac{1}{\varepsilon})$ if one only wishes to refute density estimation (or equivalently uniform estimation, see Section 6), since the assumption $\rho > \Omega(\frac{\log(1/\rho)}{\log(n)})$ can be removed in this simpler case.

## 5. Detailed proofs of Section 3.2

In this section, we show that weak density estimation is necessary but not sufficient to PAC-learn. We start with necessity.

**Proposition 34** *Let $(\mathscr{P}, X, H)$ be a class that is PAC-learnable in $n_{PAC}(\varepsilon, \delta)$ samples. Then $(\mathscr{P}, X, H)$ has a weak density estimator on*

$$n_{WDE}(\varepsilon, \delta) \leq O\left(n_{PAC}(\varepsilon/3, \delta) + \frac{\log(\Pi_H(n_{PAC}(\varepsilon/3, \delta))) + \log(1/\delta)}{\varepsilon^2}\right)$$

*samples.*

**Proof** The proof appeals to the non-uniform covering technique of Hopkins et al. (2024). In particular, draw an unlabeled sample $T$ of size $n_{PAC}(\varepsilon/3, \delta)$ and run a PAC-learner $\mathcal{L}$ across all possible labelings of $T$ to generate the set

$$C_T \coloneqq \{\mathcal{L}(T, h(T)) : h \in H\}.$$

HKLM (Hopkins et al., 2024, Lemma 5.2) observed $C_T$ non-uniformly covers $H$ in the following sense:

$$\forall h \in H : \Pr_T[\exists h' \in C_T : d_D(h, h') < \varepsilon/3] > 1 - \delta.$$

Our WDE relies on a refinement of this fact implicit in their proof. Define a mapping $c_T : H \to C_T$ by

$$c_T(h) \coloneqq \mathcal{L}(T, h(T)),$$

and observe that since $\mathcal{L}$ is an $(\varepsilon/3, \delta)$-PAC-learner, for any *fixed* $h \in H$, $c_T(h)$ is close to $h$ with high probability:

$$\forall h \in H : \Pr_T[d_D(h, c_T(h)) > \varepsilon/3] < \delta.$$

To build our WDE, this means we can simply estimate distances on $C_T$, and extend to $H$ via $c_T$.

Formally, draw a fresh sample $T'$ of $O(\frac{\log|C_T| + \log(1/\delta)}{\varepsilon^2})$ unlabeled examples. By Chernoff and Union bounds, the empirical estimator $\hat{d}_T(h, h') \coloneqq \frac{1}{|T|} \sum_{x \in T} \mathbf{1}\{h(x) \neq h'(x)\}$ is close to the true distance metric $d_D$ with high probability across elements of $C_T$:

$$\Pr_{T'}\left[\exists h, h' \in C_T : |\hat{d}_{T'}(h, h') - d_D(h, h')| > \varepsilon/3\right] < \delta.$$

We can extend $\hat{d}_{T'}$ to an approximate estimator over all of $H$ by defining for all $h, h' \in H$:

$$\tilde{d}_{T,T'}(h, h') \coloneqq \hat{d}_{T'}(c_T(h), c_T(h')).$$

We argue this approximate metric is accurate across all hypotheses which are close to their image under $c_T$. With this in mind, define

$$G_{(T,T'),D} \coloneqq \{g \in H : d_D(g, c_T(g)) \leq \varepsilon/3\}.$$

Assuming our empirical estimates on $C_T$ are good, for any $g, g' \in G_{(T,T'),D}$ we can write:

$$
\begin{aligned}
|\tilde{d}_{T,T'}(g, g') - d_D(g, g')| &= |\hat{d}_{T'}(c_T(g), c_T(g')) - d_D(g, g')| \\
&\leq |\hat{d}_{T'}(c_T(g), c_T(g')) - d_D(c_T(g), c_T(g'))| + d_D(g, c_T(g)) + d_D(g', c_T(g')) \\
&\leq \varepsilon,
\end{aligned}
$$

where the first inequality follows from noting that $d_D(g, g') = \Pr_{x \sim D}[g(x) \neq g'(x)]$ and $d_D(c_T(g), c_T(g')) = \Pr_{x \sim D}[c_T(g)(x) \neq c_T(g')(x)]$ can only differ on elements $x$ where $g(x) \neq c_T(g)(x)$ or $g'(x) \neq c_T(g')(x)$.

Putting everything together, we have that each $h \in H$ lies in $G_{(T,T'),D}$ with probability at least $1 - \delta$, and that all estimates in $G_{(T,T'),D}$ are $\varepsilon$-accurate with probability at least $1 - \delta$ giving the desired WDE. ∎

Given the closeness of weak and intermediate density estimation, one might hope there is some way to amplify weak estimation to a PAC-learner. Unfortunately, the main barrier in modifying the proof (lacking knowledge of the well-estimated set) turns out to be inherent, and we are able to give a very strong refutation of WDE's sufficiency: even a *perfect* weak estimator (i.e. one with error $\varepsilon = 0$) isn't sufficient to PAC-learn.

**Theorem 35** *There exists a class $(\mathscr{P}, X, H)$ with metric entropy $m(\varepsilon) \leq O(2^{1/\varepsilon})$ and a WDE with*

$$n_{WDE}(0, \delta) \leq O\left(\log(1/\delta)\right)$$

*samples, but is not PAC-learnable.*

We first describe the construction, which is similar in spirit to the UBME counterexample of Dudley et al. (1994). Let $X = \{0, 1, 2\}^{\mathbb{N}}$, and $\{\epsilon_i\}_{i \in \mathbb{N}}$ be the sequence $\varepsilon_i := \frac{1}{\log(i+255)}$. For each $i \in \mathbb{N}$, let $f_i$ denote the indicator that $x_i = 0$:

$$f_i(x) = \begin{cases} 1 & x_i = 0 \\ 0 & \text{otherwise,} \end{cases}$$

and define $D_i$ to be the distribution over $x \in X$ where the $i$th coordinate is chosen from $\{0, 1\}$ uniformly at random

$$\Pr[x_i = 0] = \Pr[x_i = 1] = \frac{1}{2},$$

and all remaining coordinates $j \neq i$ are drawn independently from the categorical distribution

$$\Pr[x_j = 0] = \Pr[x_j = 1] = \epsilon_j, \text{ and } \Pr[x_j = 2] = 1 - 2\epsilon_j.$$

Finally, define $\mathscr{P} = \{D_i\}_{i \in \mathbb{N}}$ and $H = \{f_i\}_{i \in \mathbb{N}}$. We first argue that $(\mathscr{P}, X, H)$ has a WDE.

**Proposition 36** $(\mathscr{P}, X, H)$ *has a weak density estimator on*

$$n_{WDE}(0, \delta) \leq O(\log(1/\delta))$$

*samples.*

**Proof** The idea is to observe that for any fixed hypotheses $f_j, f_k$, the distance $d_{D_i}(f_j, f_k)$ depends only on whether $i \in \{j, k\}$. Further, fixing $D_i$ for any particular $\ell$ it is easy to check whether $\ell = i$ by looking at the empirical mean of $f_\ell$. This allows us to define an estimator just based on empirical means which is correct across any two well-estimated hypotheses.

Formally, observe that any distribution $D_i$ and hypotheses $f_j, f_k$ where $i, j, k$ are distinct satisfy:

$$\begin{aligned} d_{D_i}(f_j, f_k) &= \Pr_{x \sim D_i} [(x_j = 0 \wedge x_k \neq 0) \vee (x_k = 0 \wedge x_j \neq 0)] \\ &= \epsilon_j(1 - \epsilon_k) + \epsilon_k(1 - \epsilon_j) \end{aligned}$$

On the other hand, hypotheses $f_i, f_j$ for $i \neq j$ satisfy:

$$\begin{aligned} d_{D_i}(f_i, f_j) &= \Pr_{x \sim D_i} [(x_i = 0 \wedge x_j \neq 0) \vee (x_j = 0 \wedge x_i \neq 0)] \\ &= \frac{1}{2}(1 - \epsilon_j) + \frac{\epsilon_j}{2} \\ &= 1/2. \end{aligned}$$

Given a sample $T$ of size $O(\log(1/\delta))$, denote the empirical density of $f_j$ under $T$ by $\mu_j^T$. We define the following approximate distance estimate on $H$:

$$\tilde{d}_T(f_j, f_k) = \begin{cases} \epsilon_j(1 - \epsilon_k) + \epsilon_k(1 - \epsilon_j) & \text{if } \mu_j^T, \mu_k^T < 1/3 \\ 1/2 & \text{otherwise.} \end{cases}$$

We argue that $\tilde{d}_T$ satisfies the conditions of a $(0, \delta)$-WDE. Let $G_{T,D}$ be the set of functions with good empirical averages:

$$G_{T,D} := \left\{ f_i : \left| \underset{D}{\mathbb{E}}[f_i] - \mu_i^T \right| < 1/6 \right\}.$$

Fix any $D_i \in \mathscr{P}$ and $f_j \in H$. First, observe that by a Chernoff bound, $f_j \in G_{T,D_i}$ with probability at least $1 - \delta$. Second, notice that since $\mu_i := \mathbb{E}_{D_i}[f_i] = 1/2$ and any other $\mu_j \leq 1/8$, for all functions $f_\ell \in G_{T,D_i}$, we are assured that $\mu_\ell^T < 1/3$ if and only if $\ell \neq i$. Since this condition exactly determines the distances between hypotheses, $\tilde{d}_T$ has zero error across $G_{T,D_i}$ as desired. $\blacksquare$

Next, we exhibit a small cover for every distribution in $(\mathscr{P}, X, H)$ based on the fact that for any fixed $D_i$, most hypotheses are close to $0$.

**Proposition 37** $(\mathscr{P}, X, H)$ *has metric entropy at most*

$$m(\varepsilon) \leq O(2^{1/\varepsilon}).$$

**Proof** Observe that for any fixed $D_i$ and $\varepsilon > 0$, the following set of hypotheses forms an $\varepsilon$-cover:

1. All hypotheses $f_j$ for which $\mathbb{E}_{D_i}[f_j] \geq \varepsilon/2$,

2. The first hypothesis $f_k$ such that $\mathbb{E}_{D_i}[f_k] \leq \varepsilon/2$,

since for any $k' > k$, $k' \neq i$ we have

$$d_{D_i}(f_k, f_{k'}) \leq \epsilon_k \leq \varepsilon$$

by construction. Moreover, by our choice of $\{\epsilon_j\}_{j \in \mathbb{N}}$, we have $k \leq O(2^{1/\varepsilon})$ which gives the desired bound on metric entropy. $\blacksquare$

Finally, we argue our class is not PAC-learnable. The proof has a similar flavor to our lower bound against intermediate density estimation: the learner cannot distinguish between ground truth and hypotheses that happen to look consistent due to a small amount of noise (here generated by the $\epsilon_i$'s).

**Proposition 38** $(\mathscr{P}, X, H)$ *is not PAC-learnable.*

**Proof** By Yao's minimax principle it suffices to find for every $t \in \mathbb{N}$ a randomized strategy for the adversary over which any deterministic learner on $t$ samples fails with at least constant probability. For each $n \in \mathbb{N}$, consider the adversary strategy that selects from the set $\{D_i, f_i\}_{i \in [n]}$ with probability:

$$\Pr[(D_i, f_i)] \propto \epsilon_i^t.$$

Denote this adversary distribution by $\mathcal{A}$ and for each $j \in [n]$, call a sample $S$ *consistent* with $(D_j, f_j)$ if for all $(x, y) \in S$, we have $x \in \text{supp}(D_j)$ and $f_j(x) = y$. For every $W \subset [n]$, let $E_W$ denote the event that the learner's sample $S$ is consistent with $(D_j, f_j)$ for all $j \in W$ and inconsistent for every $j \in [n] \setminus W$. The idea is to argue that when $S$ is consistent with multiple coordinates, the learner cannot distinguish the ground truth and fails with constant probability, and further that this almost always occurs for large enough $n$. More formally, we will show the following claims:

1. If $E_W$ holds for any $|W| > 1$, the learner must have constant error with probability $1/2$,

2. With probability at least $1/2$, $E_W$ holds for some $|W| > 1$.

Since the learner's probability of success can be broken into conditional probabilities over $E_W$ (which exactly partition the sample space), the result then follows.

To prove the first claim, we show that, conditioned on $E_W$, the posterior probability of any adversary choice $j \in W$ is uniform. To see this, first observe that fixing the choice of $(D_j, f_j)$, a coordinate $i \neq j$ of our sample $S$ is consistent if and only if every example $(x, y) \in S$ satisfies $x_i = x_j$, which occurs with probability $\epsilon_i^t$. Then by Bayes' rule for any $j \in W$ we can write:

$$\Pr_{\mathcal{A},S}[(D_j, f_j)|E_W] \propto \Pr_{\mathcal{A},S}[E_W|(D_j, f_j)] \Pr_{\mathcal{A}}[(D_j, f_j)]$$

$$\propto \left(\prod_{i \notin W} (1 - \epsilon_i^t)\right) \cdot \left(\prod_{i \in W \setminus \{j\}} \epsilon_i^t\right) \cdot \epsilon_j^t$$

$$= \left(\prod_{i \notin W} (1 - \epsilon_i^t)\right) \cdot \left(\prod_{i \in W} \epsilon_i^t\right)$$

which is independent of $j$.

We want to show that conditioned on $E_W$ for $|W| > 1$, any learner has constant error with probability at least $1/2$. By construction, it is enough to argue that any learner fails to select the adversary's choice $f_i$ itself with probability at least $1/2$ (since $f_i$ has average $1/2$ and all other options have average at most $1/4$, any other choice incurs constant error). Since the posterior over possible ground truths $f_j$ is uniform for $j \in W$, any strategy correctly outputs $f_i$ with probability at most $\frac{1}{|W|} \leq \frac{1}{2}$, which gives the desired result.

It is left to show that some $E_W$ holds for $|W| > 1$ with high probability. Given a fixed choice of $D_i$, observe the probability $|W| = 1$ is exactly

$$\Pr[E_W : |W| = 1] = \prod_{j \in [n] \setminus \{i\}} (1 - \epsilon_j^t)$$

$$\leq \prod_{j=1}^{n-1} (1 - \epsilon_j^t)$$

$$\leq e^{-\sum_{j=1}^{n-1} \epsilon_j^t}.$$

Since we chose $\epsilon_j = \frac{1}{\log(j+255)}$ to decay sufficiently slowly, the sum in the exponent diverges. Thus for any fixed $t$ there is a sufficiently large $n$ for which $\Pr[E_W : |W| = 1] \leq 1/2$ which completes the proof. ∎

# 6. Detailed proofs of Section 3.3

In this section, we give a detailed comparison between our notions of density estimation and classical notions in learning theory such as uniform estimation, uniform convergence, and UBME. This allows us to give a direct comparison of our new conditions to prior work on the distribution-family model.

### 6.1. Density vs. Uniform Estimation

We first discuss the close connection between density estimation and uniform estimation. Intuitively, it is clear the notions are intimately related. In fact, they essentially only differ in their quantification: the former promises that with a large enough sample from any fixed $D \in \mathscr{P}$, we can estimate $d_D(h, h')$ for all pairs $h, h' \in H$, while the latter promises that with a large enough sample from any fixed *pair* $(D, h)$, we can estimate $d_D(h, h')$ for all $h' \in H$.

With this in mind, one might hope to show density and uniform estimation are equivalent. In general this cannot be true since uniform estimation implies the class has uniformly bounded metric entropy, while density estimation does not (any fixed-distribution class has a trivial density estimator!). We show this is the only barrier: assuming UBME, density and uniform estimation are indeed equivalent.

**Theorem 39** *If $(\mathscr{P}, X, H)$ has a uniform estimator on $n_{UE}(\varepsilon, \delta)$ samples, then it has a density estimator on*

$$n_{DE}(\varepsilon, \delta) \leq O\left(n_{UE}(\varepsilon', \delta') + \frac{\log(\Pi_H(n_{UE}(\varepsilon', \delta'))) + \log(1/\delta)}{\varepsilon^2}\right)$$

*samples, where $\varepsilon' = O(\varepsilon)$, $\delta' = O(\frac{\delta}{\Pi_H(n_{UE}(\varepsilon/4, \delta/2))})$.*

*Conversely, if $(\mathscr{P}, X, H)$ has a density estimator on $n_{DE}(\varepsilon, \delta)$ samples and has metric entropy $m(\varepsilon)$, then it has a uniform estimator on*

$$n_{UE}(\varepsilon, \delta) \leq O\left(n_{DE}(\varepsilon/4, \delta/2) + \frac{\log(m(\varepsilon/4)) + \log(1/\delta)}{\varepsilon^2}\right)$$

*samples.*

The proof of Theorem 39 revolves (implicitly) around the fact that under UBME, both density and uniform estimation are equivalent to the ability to build a (bounded) uniform $\varepsilon$-cover $C$ and a corresponding *covering map*, that is a function $c : H \to C$ such that:

$$\forall h \in H : d_D(c(h), h) \leq \varepsilon,$$

where $D$ is the adversary's choice of distribution. We call $(C, c)$ a *covering-map pair*. This type of structure is useful since it allows us to apply Yatracos (1985)'s trick of extending empirical estimates on $C$ (which is finite), to good estimates on all of $H$ via the covering map. With this in mind we prove the backward direction of Theorem 39 which follows similarly to Theorem 35, replacing the non-uniform cover with a true covering-map pair.

**Proof** [Proof of Theorem 39: DE+UBME $\implies$ UE] Draw a sample $T$ of size $n_{DE}(\varepsilon/4, \delta/2)$ and run the density estimator to get a distribution $D' \in \mathscr{P}$ such that

$$\Pr_T[TV_{H\Delta H}(D, D') > \varepsilon/4] < \delta/2.$$

Since $(\mathscr{P}, X, H)$ has bounded metric entropy, the distribution $D'$ has a bounded $(\varepsilon/4)$-covering-map pair $(C_{D'}, c_{D'})$ with $|C_{D'}| \leq m(\varepsilon/4)$. If it is truly the case that $TV_{H\Delta H}(D', D) \leq \varepsilon/4$, then $(C_{D'}, c_{D'})$ is also an $(\varepsilon/2)$-covering-map pair for $(D, X, H)$. To build a uniform estimator, we lift empirical estimates from $C_{D'}$ to all of $H$ using the covering map $c_{D'}$.

More formally, draw a sample $S$ of $O(\frac{\log |C_{D'}| + \log(1/\delta)}{\varepsilon^2})$ fresh labeled examples to estimate the error of every hypothesis in $C_{D'}$. We define our estimator over $H$ as

$$\mathcal{E}_S(h) := \mathrm{err}_S(c_{D'}(h)).$$

So long as $(C_{D'}, c_{D'})$ is truly an $(\varepsilon/2)$-covering-map pair under $D$, the error of $h$ and $c(h)$ can differ by at most $\varepsilon/2$. Thus if the empirical errors on $C_{D'}$ are accurate up to $\varepsilon/2$, $\mathcal{E}_S$ is within $\varepsilon$ of the true risk on all hypotheses by construction. By a Chernoff and Union bound this latter guarantee also holds with probability at least $1 - \delta/2$, which completes the result. ∎

To prove the forward direction of Theorem 39, we will need the following related lemma of Hopkins et al. (2024) that shows how to build a finite cover of any learnable class.

**Lemma 40 (Learnable Classes are Coverable (Hopkins et al., 2024, Theorem G.9))** *For any $\varepsilon, \delta > 0$ and PAC-learnable class $(\mathscr{P}, X, H)$, there is an algorithm using $n_{PAC}(\varepsilon', \delta')$ samples that constructs an $\varepsilon$-cover of $H$ of size $\Pi_H(n_{PAC}(\varepsilon', \delta'))$ with probability at least $1 - \delta$ where $\varepsilon' = O(\varepsilon)$ and $\delta' = O(\frac{\delta}{\Pi_H(n_{PAC}(\varepsilon/2, 1/2))})$.*

Note that this algorithm does *not* produce a corresponding covering map (indeed this would imply necessity of density estimation, which we refuted in Section 3.1). In this case, we will use the uniform estimator to build a corresponding covering map, from which it is then easy to deduce a density estimate.

**Proof** [Proof of Theorem 39: UE $\implies$ DE+UBME] First, observe that any $(\varepsilon/2, \delta)$-uniform estimator immediately implies a PAC-learner by taking any minimizer of the output estimator $\mathcal{E}_S(\cdot)$ on $H$. By Lemma 40, this means we can build an $(\varepsilon/16)$-cover $C$ of size $\Pi_H(n_{\mathrm{UE}}(\varepsilon', \delta'))$ with probability at least $1 - \delta/3$ using $n_{\mathrm{UE}}(\varepsilon', \delta')$ labeled samples.

We now use the uniform estimator to build $C$'s associated covering map. In particular, draw an unlabeled sample $T$ of size $n_{\mathrm{UE}}(\frac{\varepsilon}{16}, \frac{\delta}{3|C|})$ from the marginal distribution $D$, and consider the family of estimators obtained by looking at all labelings of $T$ across our cover $C$:

$$\{\mathcal{E}_h\}_{h \in C}, \quad \mathcal{E}_h := \mathcal{E}_{(T, h(T))}.$$

We define our covering map by sending any $h \in H$ to its closest element in $C$ by these estimates:

$$c(h) := \mathrm{argmin}_{h' \in C} \mathcal{E}_{h'}(h),$$

breaking ties arbitrarily. Notice that as long as no run of the estimator fails (i.e. each $\mathcal{E}_{h'}$ has valid estimates within $\varepsilon/16$) and $C$ is indeed an $\varepsilon/16$-cover, then for every $h \in H$:

1. There exists $h' \in C$ such that
$$\mathcal{E}_{h'}(h) \leq \frac{\varepsilon}{8},$$

2. Every $h' \in C$ such that $\mathcal{E}_{h'}(h) \leq \frac{\varepsilon}{8}$ satisfies
$$d_D(h', h) \leq \frac{3\varepsilon}{16}.$$

Thus $(C, c)$ is an $\frac{3\varepsilon}{16}$-covering-map pair. Further, these conditions hold except with probability $2\delta/3$.

Now that we have identified a covering-map pair, we can empirically estimate distances on $C$ and extend to $H$ via $c$. Draw a sample $T'$ of $O(\frac{\log|C| + \log(1/\delta)}{\varepsilon^2})$ fresh unlabeled examples to directly estimate the distance between all hypotheses in $C$. By Chernoff and Union bounds, we have that all empirical distance estimates on $C$ are accurate up to $\varepsilon/8$ with probability at least $1 - \delta/3$. We extend these estimates to all of $H$ via the covering map similar to our approach in Proposition 34:

$$\tilde{d}_{T'}(h, h') := \hat{d}_{T'}(c(h), c(h')).$$

Since $(C, c)$ is a $\frac{3\varepsilon}{16}$-covering-map pair and distances in $C$ are well-estimated:

$$\forall h, h' \in H : |\tilde{d}_{T'}(h, h') - d_D(h, h')| \leq \varepsilon/2.$$

Finally, output any $D' \in \mathscr{P}$ approximately satisfying the estimates for all pairs $h, h' \in H$:

$$|d_{D'}(h, h') - \tilde{d}_{T'}(h, h')| \leq \varepsilon/2.$$

Such a distribution is guaranteed to exist by validity of the estimates, and satisfies $TV_{H\Delta H}(D, D') \leq \varepsilon$ by construction. Union bounding over all events, the algorithm succeeds with probability at least $1 - \delta$ which completes the proof. ∎

We note that (as stated in Theorem 7) it is possible to give an equivalence between intermediate density estimation and a corresponding relaxed form of uniform estimation through exactly the same argument (we omit the details). This is likely not the case for weak density estimation, where the situation is more subtle. In particular the natural notion of weak uniform estimation (where the error of most hypotheses must be well-estimated, but the learner does not know which ones) is essentially trivial for any class by Chernoff. On the other hand, weak density estimation asks for more than just a good estimate on most pairs (which would follow similarly), but rather for the existence of a large subclass where *all* estimates are good. We leave the characterization of classes satisfying this latter property as a problem for future work.

Since uniform estimation characterizes learnability in the distribution free setting,[11] Theorem 39 immediately implies that density estimation is an equivalent characterization of the traditional PAC model.

**Corollary 41** *In the distribution-free setting, $(X, H)$ is PAC-learnable if and only if it has UBME and a density estimator.*

**Proof** If $(X, H)$ is PAC-learnable, then it satisfies uniform convergence (Vapnik and Chervonenkis, 1974; Blumer et al., 1989) which implies the existence of a density estimator by Theorem 39. If $(X, H)$ has a density estimator and has UBME, then it is PAC-learnable by Proposition 29. ∎

We remark that in the distribution-free case, one does not have to factor through uniform estimation, and could instead use a more classical approach by arguing $H\Delta H$ has finite VC and therefore satisfies fast convergence of empirical densities in $H\Delta H$-distance (see e.g. (Anthony et al., 1999, Theorem 4.9)). This approach fails in our setting since $H\Delta H$ need not have finite VC.

---

11. Unlike the general setting, uniform estimation and uniform convergence are equivalent in the distribution-free case.

### 6.2. Uniform Convergence and UBME

We now discuss the relation between density estimation and prior conditions considered in the distribution-family model: uniform convergence (for sufficiency) and uniformly bounded metric entropy (for necessity). First, we show that density estimation strictly contains uniform convergence.

**Proposition 42** *In the distribution-family model:*

$$\textit{Uniform Convergence} \subsetneq \textit{DE + UBME.}$$

**Proof** Containment is immediate from the characterization of DE + UBME as uniform estimation (Theorem 39). Strictness follows from Benedek and Itai (1991)'s example of a learnable class that fails uniform convergence. Let $D$ be the uniform distribution over $[0, 1]$, and let $H$ consist of all sets of finite support and the all 1's function. It is easy to see uniform convergence fails, since choosing $h = \mathbf{1}$, for any finite sample $T$ there is always a completely consistent hypothesis with error 1, namely the indicator $\mathbf{1}_T$. On the other hand, since $\mathscr{P}$ is just a single distribution, density estimation is trivial. ∎

Finally, we discuss the connection between weak density estimation and UBME.

**Proposition 43** *Any class $(\mathscr{P}, X, H)$ with metric entropy $m(\varepsilon)$ has a weak density estimator on*

$$n_{\textit{WDE}}(\varepsilon, \delta) \leq O\left(\frac{\log(m(\varepsilon/8)) + \log(1/\delta)}{\varepsilon^2}\right)$$

*samples.*

**Proof** We argue it is sufficient to simply output the empirical distance estimator on an unlabeled sample $T$ of size $O(\frac{\log(m(\varepsilon/8)) + \log(1/\delta)}{\varepsilon^2})$. Fix any $D \in \mathscr{P}$, and let $(C, c)$ denote an $(\varepsilon/8)$-covering-map pair promised by UBME. Note that $(C, c)$ is not known to the learner, and is used only in the analysis.

By Chernoff and Union bounds, we have with probability at least $1 - \delta$ that the empirical distance estimator $\hat{d}_T$ is within $\varepsilon/8$ of its true value for all pairs in $C$. Now consider the set $G_{T,D} \subset H$ of hypotheses whose distance to their cover representative is well estimated:

$$G_{T,D} := \left\{ h \in H : |\hat{d}_T(h, c(h)) - d_D(h, c(h))| \leq \varepsilon/8 \right\}.$$

For any two elements $h, h' \in G_{T,D}$, we have

1. The distances $d_D(h, h')$ and $d_D(c(h), c(h'))$ are close:

$$|d_D(h, h') - d_D(c(h), c(h'))| \leq \varepsilon/4$$

2. The empirical distances $\hat{d}_T(h, h')$ and $\hat{d}_T(c(h), c(h'))$ are close

$$|\hat{d}_T(h, h') - \hat{d}_T(c(h), c(h'))| \leq \varepsilon/2$$

since $h\Delta h'$ and $c(h)\Delta c(h')$ can only differ on $h\Delta c(h) \cup h'\Delta c(h')$, both of which have true measure at most $\varepsilon/8$ and empirical measure at most $\varepsilon/4$ by assumption. Thus as long as the empirical distance estimates on $C$ are indeed accurate up to $\varepsilon/8$, we have

$$|\hat{d}_T(h, h') - d_D(h, h')| \leq |\hat{d}_T(c(h), c(h')) - d_D(c(h), c(h'))| + 3\varepsilon/4$$
$$\leq \varepsilon$$

as desired. Since any fixed $h \in H$ lies in $G_{T,D}$ with probability at least $1 - \delta$ by a Chernoff bound and distances in $C$ are also well estimated with probability at least $1 - \delta$, this gives the desired estimator. ■

Note that if one is not interested in relative sample complexities, this subsumes Proposition 34 since any PAC-learnable class has UBME (Benedek and Itai, 1991). Proposition 43 also implies that the counter-example of Dudley et al. (1994) shows weak estimation fails sufficiency as well. However, our construction in the previous section gives a stronger separation since the above approach is inherently lossy and cannot handle $\varepsilon = 0$.

## Acknowledgments

We thank Shay Moran for many interesting discussions surrounding the topic of this work and anonymous reviewers for pointing out connections with density estimation and suggesting improved naming conventions.

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
