# OpenReview forum: "Do PAC-Learners Learn the Marginal Distribution?"
_algorithmiclearningtheory.org/ALT/2025/Conference — ALT 2025_

### Official Review · Reviewer_k7bZ · 2024-11-08

**Rating:** 7
**Confidence:** 3

**Review:**

Summary:

In standard PAC learning, the adversary can select any distribution. PAC-learnability is then known to be equivalent to uniform convergence, and also to density estimation (the task of estimating the true distribution in a total variation metric that is restricted to a set of events determined by the hypothesis class).

The paper studies a less adversarial "Distribution Family" version of PAC learning where the adversary is restricted to select a distribution from a known set of distributions. It shows that:
  a) Distribution family PAC-learnability is harder than weak density estimation, but easier than intermediate density estimation.
  b) If the distribution family has uniformly bounded metric entropy, then a distribution family variant of density estimation is equivalent to uniform estimation.

Here, weak and intermediate density estimation are learning tasks in which the goal is to estimate the probability that two classifiers make different predictions, for most pairs of classifiers in the hypothesis class; they differ in the details of defining 'most'. Uniform estimation is the task of uniformly estimating the risk for all hypotheses in the hypothesis class.

Significance of the results:

PAC learning is a central topic in learning theory, and its relaxation to a distribution family is natural and practically significant. As far as I am aware, all main results are new. They give interesting fundamental insights into the relations between various notions of distribution family learnability. This work will therefore clearly be of interest to the ALT community.

A limitation of the work is that it is not shown whether the derived relations between sample complexities are optimal. But this would also not be expected in a first work to establish the relations.

Correctness: I have selectively checked part of the proofs, and found no mistakes.

Question for the authors:

Q. I got the impression that intermediate and weak density estimation are new learning tasks. Is this correct or have they been considered before? And the distribution-family versions of density estimation and uniform estimation?

Presentation and clarity: the paper is generally well-written. I have a few minor suggestions to improve the clarity and presentation, which are described below.

All in all, I believe this is interesting work that gives new insights into fundamental learning tasks, and I recommend it for acceptance.

High-level suggestions to improve the presentation:

* I strongly feel that the purpose of 'Theorem', 'Definition' and 'Proposition' should remain to identify formal statements, as it is throughout mathematics. So even though I really like the informal paraphrasing of several theorems and definitions in the introduction, the authors should avoid labeling them with self-contradictory phrases like 'informal theorem' or 'informal definition'. This is especially problematic in cases like Proposition 7, where the 'informal' version is not just paraphrasing the formal result, but it is also leaving out essential conditions that are necessary for it to hold.

* The structure of the paper is not clear at the start. I would suggest to add a brief outline at the end of the introduction that summarizes the technical contributions and discusses the structure of the paper. I would further suggest to rename section 3 to 'Main Results'. Then, for i=1,2,3, I would suggest to rename Section 3+i to "Detailed Proofs for Section 3.i".

* The long and short versions of the proofs are a bit repetitive, with many sentences repeated verbatim. This is not the most elegant. If possible, it would nice to break things up in a way that avoids too much repetition. Otherwise, the current form is acceptable as is.

Minor comments:

* The abstract uses cal{H} whereas the rest of the paper uses H
* Definition 2: "exists" -> "exist"
* Please define H Delta H. The description in footnote 1 is not clear.
* Footnote 2: uniformly bounded metric entropy of what?
* p.5: In definition of growth function, define H_{|S} (even though it is well-known)
* Definition 15: state that G_{T,D} is random. Its existence should also be stated before '1.' because '2.' also refers to it.
* p.6: "classical problemS in statistics"
* Section 3: please start section with short overview of its structure
* Proposition 21: recall that Pi_H is the growth function (it took me a while to find its definition)
* Theorem 27: typo in delta' = O(delta/Pi_H(...)): the part in place of ... should be an integer, probably you meant n_UE(eps/4, delta/2).
* A conclusion with suggestions for future work at the end of paper could make a nice addition.

**Paper Award:**

No

---

> ### Author Response · Authors · 2024-11-20
> **Author Response**
>
> We thank the reviewer for their careful reading and evaluation of our work, as well as their presentation suggestions.
>
> *Regarding the reviewers question:*
>
> 1. Intermediate and weak density estimation are new learning tasks to the best of our knowledge.
>
>
> 2. Uniform convergence has been studied in the distribution family model based on expected bounds on the growth function (some variant of this was even done in initial works of Vapnik and Chervonenkis), but again to our knowledge uniform *estimation* (which is more appropriate in the setting due to Benedek and Itai’s classic counter-example to uniform convergence even over a single fixed distribution) has not been studied in the setting
>
>
> 3. Classical density estimation in statistics typically considers a restricted distribution family but takes H to be all possible functions (i.e. just measures standard TV distance). Most relevant works in the learning literature are in distribution learning,(e.g. efficiently learning mixture of gaussians), where again H is all possible functions. However, there are cases in such works where considering density estimation with respect to a fixed VC class is useful as an ancillary tool (see e.g. Lemma 6 in https://arxiv.org/pdf/1605.08188). In this sense the setting is not really new, though to our knowledge it has not been explicitly studied in its own right, e.g. when H is not a VC class as is the interesting setting in our work.
>
>
> *Regarding the reviewers presentation suggestions:*
>
> We thank the reviewer again for their careful reading and will make most of the suggested adjustments.
>
> We respectfully disagree with the reviewer on the use of informal theorems in the introduction. This is common practice in theoretical computer science, and we believe it is a useful tool to convey the main ideas and results to the reader without requiring full technical background. To ensure there isn’t confusion from informality, we will make sure every such result hyperlinks to the fully formal version (which we included for most informal statements with the exception of Theorem 6). We will also try to make the informal version of Proposition 7 more clear.

---

> > ### Comment · Reviewer_k7bZ · 2024-11-26
> > **Re: Author Response**
> >
> > Dear authors,
> >
> > Thank you for your clarifications.
> >
> > Regarding the use of "informal theorems": in the end this is a stylistic issue, which should not affect acceptance. So since I have not managed to convince you, it is your prerogative to keep them.

---

### Official Review · Reviewer_vK1Y · 2024-11-09

**Rating:** 7
**Confidence:** 4

**Review:**

The fundamental theorem of PAC learning states that the learnability of some class H depends on the uniform convergence, in the sense that the empirical mean of a sufficiently large sample estimates the error of every hypothesis $h \in H$. However, this is a general distribution-free notion and works for all adversarial settings. Moreover, this is equivalent to the problem of density estimation and the problem of uniform convergence. Since the result is so general, the sample complexity can often be improved in practice with additional information This paper studied this problem in a restricted adversarial setting.

The paper defines two notions of density estimation: (i) Intermediate Density estimation (IDE) (Definition 2) and (ii) Weak Density Estimation (WDE) (Definition 4). In the problem of IDE, instead of learning all hypotheses in H, the goal is to have the guarantee for a subset $G \subset H $ which is known to the learner. For the problem of WDE, the learner does not need to know G.

The main result of the paper (see Theorem 6, Section 1) shows that the three notions of WDE, PAC learning and IDE are not same akin to the fundamental theorem of PAC learning (see Theorem 1). Instead, PAC learning is sandwiched between WDE and IDE. Additionally, the authors proved that IDE is equivalent to intermediate uniform estimator (Proposition 7) under the assumption of Uniformly Bounded Metric Entropy (see Definition 12, Section 2).

To prove the second part of Theorem 6 (PAC learning is contained in IDE, Section 4), the authors first show that if a hypothesis class H has bounded metric entropy and has an intermediate density estimator that requires a bounded number of samples, then H has an efficient PAC learner (Proposition 29, Section 4). Additionally, they prove that the converse does not hold. In particular, there exists a concept class that can be PAC learned efficiently, however, no efficient intermediate density estimation of H is possible (Theorem 30). The lower bound follows via Yao's minimax method, and the upper bound follows from the assumption of bounded metric entropy.

To prove the first part of Theorem 6 (WDE is contained in PAC learning, Section 5), similar to the previous case, the authors first prove that if a hypothesis class H is PAC learnable, then it also has a Weak density Estimator (Proposition 34, Section 5).  Additionally, they prove that there exists a class for which efficient WDE exists, which is not PAC learnable (Theorem 35). The upper bound uses approaches from the work of Hopkins et al. (TheoretiCS 24).

Finally, in Section 6, the authors show an equivalence between uniform estimation (Definition 26, Section 3.3) and density estimation when H has bounded metric entropy (Theorem 39, Section 6). In Section 6.2, the authors prove that uniform convergence is strictly contained in Density estimation with bounded metric entropy (Proposition 42) and every class with bounded metric entropy has a weak density estimator (Proposition 43). The authors also present an overview of the technical results in Section 3.

Overall, I like the set of results and believe they are interesting for the ALT community. I support accepting the paper.

**Paper Award:**

No

---

> ### Author Response · Authors · 2024-11-20
> **Author Response**
>
> We thank the reviewer for their careful reading and evaluation of our work.

---

### Official Review · Reviewer_FpMi · 2024-11-11

**Rating:** 7
**Confidence:** 3

**Review:**

### Summary
This paper studies the connection between
* PAC learning a (Booelan) concept class $\mathcal{H}$ over input space $X$ _under distributional assumptions_, and
* the problem of _density estimation_, where the goal is to learn the marginal distribution with respect to events in $\mathcal{H}$.

The distributional assumption is modeled by requiring that the marginal distribution over inputs is some distribution $D$ that belongs to a known family $\mathscr{P}$. It is said that $(\mathscr{P}, X, H)$ is PAC-learnable with sample complexity $n(\varepsilon, \delta)$ if there exists an algorithm which for every $D \in \mathscr{P}$ and $h \in H$ outputs a predictor $h'$ such that $\Pr_{x \sim D}[h'(x) \ne h(x)] \le \varepsilon$ with probability at least $1 - \delta$.

Formally, the problem of density estimation is, given an input dataset $T \sim D^n$, to output a density $D'$ such that $TV_{H \Delta H}(D', D) \le \varepsilon$ with probability at least $1 - \delta$, where $TV_{H \Delta H}(D', D) := \sup_{h, h' \in H} |d_{D'}(h, h') - d_D(h, h')|$ where $d_D(h, h') := \Pr_{x \sim D}[h(x) \ne h'(x)]$.\
This paper studies two different weakenings of this definition, in both of which instead of outputting a density $D'$, it is only required to output a distance estimator $\tilde{d}_T : H \times H \to \mathbb{R}^+$:
* _Intermediate Density Estimation (IDE)_: It is required that there exists a set $G\_T$ (that depends on the dataset $T$ such that for all $g, g' in G\_T$ it holds that $|\tilde{d}_T(g, g') - d_D(g, g')| \le \varepsilon$ holds with probability at least $1-\delta$, and that any fixed hypothesis $h \in H$ belongs to $G_T$ with probability at least $1-\delta$.
* _Weak Density Estimation (WDE)_: The same as IDE, but where the set $G\_T$ may be unknown to the learner (hence, denoted $G_{T, D}$).

The main result of the paper is to show that PAC-learning lies strictly between the notions of IDE and WDE. Namely,
* any $(\mathscr{P}, X, H)$ that admits IDE, also admits PAC-learning, and moreover, this is strict, in that there exists an example that admits PAC-learning, but does not admit IDE.
* any $(\mathscr{P}, X, H)$ that admits PAC-learning also admits WDE, and moreover, this is strict, in that there exists an example that admits WDE, but does not admit PAC learning.

Finally an equivalence is shown between $(\mathscr{P}, X, H)$ satisfying the _uniform estimation property_ and admitting density estimation (although the converse also requires uniformly bounding the metric entropy).

### Relevance and Recommendation
The paper provides a new insight on PAC learning under distributional assumptions, and provides a novel understanding in terms of _intermediate_ and _weak_ density estimation. I feel this paper is likely to inspire follow up work. So I recommend acceptance.

### Questions for Authors
One thing that was not clear to me was whether the $\tilde{d}_T : H \times H \to \mathbb{R}^+$ as constructed in this work always corresponds to an actual density function over $X$? (I understand this is not relevant to the proofs in the paper, but was just curious for my own sake.)

### Minor Comments

(Page 7) Proposition 16: Should $m(\varepsilon)$ be in fact $m_{\mathrm{UB}}(\varepsilon)$ ?

**Paper Award:**

No

---

> ### Author Response · Authors · 2024-11-20
> **Author Response**
>
> We thank the reviewer for their careful reading and evaluation of our work. Regarding whether the estimator d_T comes from a true density function: this isn’t necessarily the case (e.g. in Prop 21), but it is always possible to construct such an estimator if one takes a number of samples scaling with the metric entropy of the class (it is then enough to take d_T to be given by the empirical measure on the sample).

---

> > ### Comment · Reviewer_FpMi · 2024-11-26
> >
> > I thank the authors for their response.
> >
> > What I meant to ask is: Is it clear that for the $\tilde{d}_T$ that is shown to exist, does there exist a probability distribution $\tilde{D}$ (continuous or not) such that $\tilde{d}\_T(g, g') = d\_{\tilde{D}}(g, g')$ ? From the notation it seems that $\tilde{d}_T : H \times H \to \mathbb{R}^+$ can be an arbitrary function.

---

> > > ### Author Response · Authors · 2024-11-27
> > > **Author Response**
> > >
> > > Thank you for the clarification.
> > >
> > > In Proposition 21, it is not clear the constructed WDE $\tilde{d}_T: H \times H \to \mathbb{R}^+$ comes from any distribution -- as you suggest it is just some arbitrary function on hypothesis pairs satisfying the desired distance estimates (albeit one that is efficiently computable given oracle access to the PAC Learner).
> > >
> > > In Proposition 28, on the other hand, the constructed WDE $d_T: H \times H \to  \mathbb{R}^+$ does come from some distribution $\tilde{D}$, i.e. is of the form $d_{\tilde{D}}(h,h')$ where $\tilde{D}$ is the uniform distribution over a large enough empirical sample.

---

### Meta-Review · Area_Chair_Cp3V · 2024-12-13

**Recommendation:** Accept
**Confidence:** 4

**Metareview:**

The paper studies to what extend the fundamental theorem of PAC learning can be extend to the setting there the adversary's choice is restricted to a distribution family. The main result is to establish two relaxed notions of density estimation, which are strictly weaker and strictly stronger than PAC learning. The paper also establishes an equivalence of PAC learning and uniform estimation, a relaxed notion of uniform convergence. The reviewers all appreciate the novelty and fundamental insights of the study and its importance to the ALT community. I concur with the reviewers' positive assessment and I think this work is likely to inspire further research in the field. Hence I recommend the paper for acceptance.

**Paper Award:**

No